# Coordination of two enhancers drives expression of olfactory trace amine-associated receptors

Aimei Fei[1,2,9], Wanqing Wu[1,2,9], Longzhi Tan [3,9], Cheng Tang[4,9], Zhengrong Xu[1,2], Xiaona Huo[4], Hongqiang Bao[1,2], Yalei Kong[1,2], Mark Johnson[5], Griffin Hartmann[5], Mustafa Talay [5], Cheng Yang[1,2], Clemens Riegler[6], Kristian J. Herrera [6], Florian Engert [6], X. Sunney Xie[3], Gilad Barnea [5], Stephen D. Liberles[7], Hui Yang [4] & Qian Li [1,2,8✉]

Olfactory sensory neurons (OSNs) are functionally defined by their expression of a unique odorant receptor (OR). Mechanisms underlying singular OR expression are well studied, and involve a massive cross-chromosomal enhancer interaction network. Trace amine-associated receptors (TAARs) form a distinct family of olfactory receptors, and here we find that mechanisms regulating *Taar* gene choice display many unique features. The epigenetic signature of *Taar* genes in TAAR OSNs is different from that in OR OSNs. We further identify that two TAAR enhancers conserved across placental mammals are absolutely required for expression of the entire *Taar* gene repertoire. Deletion of either enhancer dramatically decreases the expression probabilities of different *Taar* genes, while deletion of both enhancers completely eliminates the TAAR OSN populations. In addition, both of the enhancers are sufficient to drive transgene expression in the partially overlapped TAAR OSNs. We also show that the TAAR enhancers operate in *cis* to regulate *Taar* gene expression. Our findings reveal a coordinated control of *Taar* gene choice in OSNs by two remote enhancers, and provide an excellent model to study molecular mechanisms underlying formation of an olfactory subsystem.

[1] Center for Brain Science of Shanghai Children's Medical Center, Shanghai Jiao Tong University School of Medicine, Shanghai, China. [2] Department of Anatomy and Physiology, Shanghai Jiao Tong University School of Medicine, Shanghai, China. [3] Department of Chemistry and Chemical Biology, Harvard University, Cambridge, MA, USA. [4] Institute of Neuroscience, State Key Laboratory of Neuroscience, Key Laboratory of Primate Neurobiology, CAS Center for Excellence in Brain Science and Intelligence Technology, Shanghai Research Center for Brain Science and Brian-Inspired Intelligence, Shanghai Institutes for Biological Sciences, Chinese Academy of Sciences, Shanghai, China. [5] Department of Neuroscience, Division of Biology and Medicine, Brown University, Providence, RI, USA. [6] Department of Molecular and Cellular Biology and Center for Brain Science, Harvard University, Cambridge, MA, USA. [7] Howard Hughes Medical Institute, Department of Cell Biology, Harvard Medical School, Boston, MA, USA. [8] Shanghai Research Center for Brain Science and Brain-Inspired Intelligence, Shanghai, China. [9] These authors contributed equally: Aimei Fei, Wanqing Wu, Longzhi Tan, Cheng Tang. ✉email: liqian@shsmu.edu.cn

In the mammalian olfactory systems, the ability to detect and discriminate a multitude of odorants relies on the expression of a wide range of receptor genes in olfactory sensory neurons (OSNs)[1]. In the main olfactory epithelium (MOE), olfactory receptor genes are mainly composed of two families of seven-transmembrane G protein-coupled receptors (GPCRs): odorant receptors (ORs)[1] and trace amine-associated receptors (TAARs)[2]. Some OSNs express other types of olfactory receptors that are not GPCRs: membrane-spanning 4-pass A receptors[3] and guanylyl cyclase D[4,5]. OSNs expressing different olfactory receptor gene families constitute distinct olfactory subsystems that detect specific categories of odorants. In addition to odorant recognition, ORs also play an instructive role in targeting the axons of OSNs into specific locations in the olfactory bulb[6–8]. Thus, correct expression of olfactory receptor genes is critical for precise translation of external odor information into the brain.

In mice, the *OR* gene family consists of ~1100 functional genes and form the largest gene family in the genome. *OR* transcription is initiated by epigenetic switch from the repressive H3K9me3/H4K20me3 (histone H3 trimethyl lysine 9 and histone H4 tri-methyl lysine 20) state to the active H3K4me3 (histone H3 tri-methyl lysine 4) state on a stochastically chosen *OR* allele[9]. This process involves an enzymatic complex with histone demethylases, including LSD1 (lysine-specific demethylase 1)[10]. Once a functional OR protein is expressed, a feedback signal is triggered to prevent the de-silencing of other *OR* genes by inhibiting the histone demethylase complex, thereby stabilizing the *OR* gene choice throughout the lifetime of each OSN[11–14]. In addition, *OR* gene choice is facilitated by multiple intergenic OR enhancers (63 in total, also known as "Greek Islands") that interact with each other to form an interchromosomal enhancer hub[15–18]. The OR enhancer hub is hypothesized to insulate a chosen active *OR* allele from the surrounding repressive heterochromatin compartment. Thus, deletion of a single OR enhancer typically affects 7–10 *OR* genes within the same *OR* clusters adjacent to the enhancer (within 200 kb). For instance, knockout of the H, P, or Lipsi enhancers decreases the expression of 7, 10, or 8 *OR* genes, respectively[15,19,20]. One exception is the J element (or called element A[21]) that exerts its function on 75 *OR* genes over ~3 megabase genomic distance[22]. Nevertheless, while the OR enhancers interact in *trans* to form the multi-enhancer hub, their restricted effects on proximal *OR* gene expression suggest that they operate in *cis*. This may be due to redundancy among the 63 enhancers in hub formation. As a result, deletion of a single or a few OR enhancers did not result in overall changes in *OR* gene expression[17].

On the other side, TAARs form a distinct olfactory receptor subfamily and are evolutionarily conserved in jawed vertebrates, including humans[23]. TAARs are distantly related to biogenic amine receptors, such as dopamine and serotonin receptors, and not related to ORs. The TAAR family is much smaller than the OR family, with 15 functional members in mouse, 17 in rat, and 6 in human[24]. In mouse, *Taar* genes are arranged in a single cluster in chromosome 10 and are numbered based on their chromosomal order, from *Taar1* to *Taar9*, with five intact *Taar7* genes (*Taar7a, Taar7b, Taar7d, Taar7e,* and *Taar7f; Taar7c* is the only *Taar* pseudogene) and three intact *Taar8* genes (*Taar8a, Taar8b,* and *Taar8c*). *Taar1* is mainly expressed in the brain, while the other *Taar* genes are mainly expressed in the dorsal zone of MOE except that *Taar6* and two members of *Taar7* (*Taar7a* and *Taar7b*) are expressed more ventrally[2,25,26]. Several TAARs respond to volatile amines some of which are ethologically relevant odors that serve as predator signals or social cues and evoke innate behaviors[27–35]. *Taar* genes do not co-express with *OR* genes, suggesting that TAARs constitute a distinct olfactory

subsystem[2]. Consistent with this notion, genetic evidences suggest distinct mechanisms of receptor choice for *Taar*s and *OR*s[25,26]. However, the nature of the regulatory mechanisms of *Taar* gene choice remains elusive.

Here, we found that the *Taar* gene cluster is decorated by different heterochromatic marks in TAAR and OR OSNs. We further searched for TAAR-specific enhancers by performing assay for transposase-accessible chromatin using sequencing (ATAC-seq) on purified TAAR OSNs. We identified two TAAR enhancers in the 200 kb *Taar* gene cluster, with TAAR enhancer 1 located between *Taar1* and *Taar2* and TAAR enhancer 2 between *Taar6* and *Taar7a*. Both enhancers are evolutionarily conserved in placental mammals, including humans. Deletion of either enhancer leads to specific abolishment or significant decrease of distinct TAAR OSN populations, suggesting that the two enhancers act in a non-redundant manner. Furthermore, the entire TAAR OSN populations are eliminated when both of the TAAR enhancers are deleted. In transgenic animals bearing the TAAR enhancers, each enhancer is capable of specifically driving reporter expression in subsets of TAAR OSNs. We next provide genetic evidence that the TAAR enhancer 1 acts in *cis*. Taken together, our study reveals two enhancers located within a single gene cluster that coordinately control the expression of an entire olfactory receptor gene subfamily.

## Results

**Enrichment of TAAR OSNs.** In order to gain genetic access to TAAR-expressing OSNs, we generated *Taar5-ires-Cre* and *Taar6-ires-Cre* knockin mouse lines (Fig. 1a), in which Cre recombinase is co-transcribed with *Taar5* and *Taar6* gene, respectively. Following transcription, Cre is independently translated from an internal ribosome entry site (IRES) sequence[36]. Each Cre line was crossed with Cre-dependent reporter lines, including *lox-L10-GFP*[37] and *lox-ZsGreen*[37], to fluorescently label the specific population of TAAR OSNs. The green fluorescent protein (GFP)- or ZsGreen-positive cells were sorted by fluorescence-activated cell sorting (FACS; Supplementary Fig. 1a) and enrichment of TAAR OSNs was verified by RNA-seq. The GFP- or ZsGreen-negative cells were also sorted to serve as control cells, approximately 70–80% of which are composed of OR-expressing OSNs. This percentage was estimated from our previous observation[38]. We indeed observed a dramatic increase in *Taar* gene expression and decrease in *OR* gene expression in sorted positive cells compared to control cells (Fig. 1b).

Next, we examined the expression of individual *Taar* genes. There are 15 functional *Taar* genes and 1 pseudogene (*Taar7c*) clustered on mouse chromosome 10 forming the single *Taar* cluster without any other annotated genes. Consistent with the previous observation that *Taar1* is mainly expressed in the brain[39], we did not detect *Taar1* expression in the sorted cells (Fig. 1c). Surprisingly, all of the other 14 functional *Taar* genes were detected at various expression levels in reporter-positive cells (Fig. 1c). We expected to obtain pure *Taar5* or *Taar6* expression as TAAR OSNs obey the "one-neuron-one-receptor" rule[2]. A parsimonious interpretation of this observation is that TAAR OSNs undergo receptor switching at much higher frequencies than OR OSNs[14]. Alternatively, transient coexpression of multiple *Taar* genes in immature TAAR OSNs revealed by two-color in situ hybridization in our previous study could also explain the phenomena[38]. Moreover, we found that *Taar4* was dominantly expressed in *Taar5+* OSNs, and that both ventral and dorsal *Taar* genes were expressed in *Taar6+* OSNs (Fig. 1c). This is consistent with a previous study analyzing coexpression of *lacZ* gene with *Taar* genes in *Taar5^{lacZ/+}* or *Taar6^{lacZ/+}* mice, where the *Taar5* or *Taar6* coding region was replaced by a *lacZ*

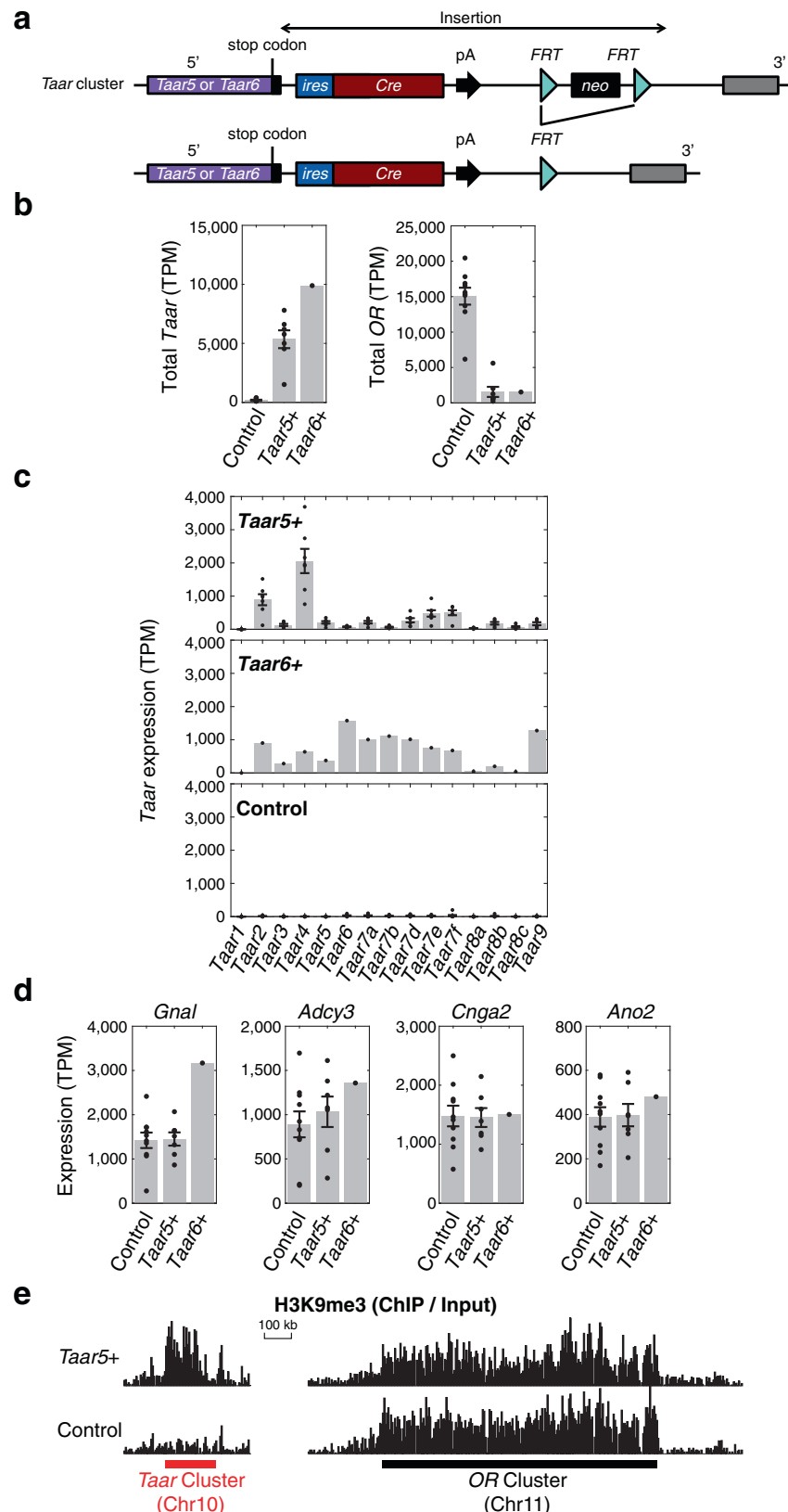

gene in one allele[40]. Nevertheless, we have successfully enriched TAAR OSNs that express a mixture of different TAAR family members. In addition, we analyzed the canonical OSN signaling proteins and chaperones, including *Gnal*, *Adcy3*, *Cnga2*, *Ano2*, and *Rtp1/2*, in enriched TAAR OSNs. We observed comparable expression levels to control cells (Fig. 1d and Supplementary

Fig. 1b), suggesting that TAAR OSNs use the same signaling pathways as OR OSNs.

**The *Taar* cluster is covered by heterochromatic marks in TAAR OSNs but not in OR OSNs.** Previous studies have shown

**Fig. 1 Enrichment of TAAR OSNs. a** Schematic illustration of the design strategy for *ires-Cre* knockin mice. DNA sequences encoding the *ires-Cre* alleles were inserted after the endogenous stop codon of *Taar5* or *Taar6* genes, along with the gene encoding neomycin (neo) resistance. The neo cassette flanked by flippase recognition target (FRT) sites was further excised by crossing to mice that express germline Flp recombinase. *Taar5-ires-Cre* or *Taar6-ires-Cre* mice were then crossed with Cre-dependent reporter lines to allow for fluorescent labeling of TAAR OSNs. **b** FACS-sorted reporter-positive (*Taar5+* or *Taar6+*) and reporter-negative (control) cells were collected for RNA-seq experiments. The values of transcripts per million (TPM) extracted for the total *Taar* and *OR* genes were plotted. **c** The TPM values of all the 15 *Taar* genes were plotted in reporter-positive (*Taar5+* or *Taar6+*) and reporter-negative (control) cells, showing that reporter-positive cells were composed of mixed TAAR OSN populations. **d** The canonical olfactory signaling molecules were expressed at similar level in reporter-positive (*Taar5+* or *Taar6+*) and reporter-negative (control) cells, including *Gnal*, *Adcy3*, *Cnga2*, and *Ano2*. **e** H3K9me3 ChIP-seq results showed that both the *Taar* cluster and the *OR* cluster were decorated by H3K9me3 in *Taar5+* OSNs. In contrast, the *Taar* cluster was devoid of H3K9me3 decoration in control cells that are mainly OR OSNs. In **b–d**, $n = 10$ for control cells, $n = 7$ for *Taar5+* reporter-positive cells, $n = 1$ for *Taar6+* reporter-positive cells. Data are presented as mean values ± SEM.

that, unlike the *OR* clusters, the *Taar* cluster is not decorated by heterochromatic silencing marks of H3K9me3, indicating a different regulatory mechanism on receptor gene choice[26]. However, the experiments were performed on the whole MOE tissue, which mostly consists of OR OSNs and contains <1% TAAR OSNs. Therefore, it is possible that H3K9me3 is excluded from the *Taar* cluster in OR OSNs and not in TAAR OSNs, but it is diluted below the level of detection in the whole MOE preparation. To test this possibility, we carried out H3K9me3 native chromatin immunoprecipitation–sequencing (ChIP-seq) analysis on purified TAAR OSNs from *Taar5-ires-Cre*; *lox-L10-GFP* mice and found that the *Taar* cluster, as well as the *OR* clusters, are actually marked with high levels of H3K9me3 modification (Fig. 1e). By contrast, in the sorted GFP-negative cells, the *Taar* cluster is devoid of H3K9me3 repressive marks, whereas H3K9me3 marks are enriched in the *OR* clusters (Fig. 1e), in agreement with the previous study[26]. Note that, limited by low cell numbers, our data does not have single-*Taar*-gene resolution. Furthermore, the *Taar* cluster are not marked by H3K9me3 in globose basal stem cells (GBCs), immediate neuronal precursors (INPs), immature OSNs (iOSNs), or mature OSNs prepared from the whole MOE according to the recent ChIP-seq data (personal communication with Stavros Lomvardas and Lisa Bashkirova; data from ref. [41]), indicating that the H3K9me3 decoration on the *Taar* cluster is absent throughout the OR OSN lineage. However, since we do not have genetic mouse lines to enrich GBC, INP, and iOSN subpopulations of the TAAR OSN lineage, we can only postulate that the H3K9me3 decoration on the *Taar* cluster arises in TAAR OSN progenitors. Together, these results suggest that *Taar* gene expression undergoes epigenetic regulation in TAAR OSNs analogous to the regulation of *OR* genes in OR OSNs and that it may also require histone demethylases, including LSD1. However, the mechanisms of *Taar* gene silencing in OR OSNs might be different from *OR* gene silencing in TAAR OSNs.

**Identification of two putative TAAR enhancers.** To further dissect the regulatory elements of *Taar* genes, we performed ATAC-seq to assay regions of open chromatin in TAAR OSNs sorted from *Taar5-ires-Cre*; *lox-L10-GFP*, *Taar5-ires-Cre*; *lox-ZsGreen*, or *Taar6-ires-Cre*; *lox-ZsGreen* mice[42]. As a control, all mature OSNs that are mainly composed of OR OSNs were sorted from *Omp-ires-GFP* mice (Fig. 2a). In total, we obtained six replicates of TAAR OSN samples and three replicates of olfactory marker protein (OMP)-positive mature OSN samples for ATAC-seq. We first examined a number of genes encoding olfactory signaling molecules, such as *Gnal*, *Adcy3*, *Cnga2*, *Ano2*, and *Rtp1/2*. Consistent with the high levels of these genes in RNA-seq data, we observed strong ATAC-seq peaks with comparable intensities in both TAAR and OMP-positive OSNs (Supplementary Fig. 2). The peaks are mostly located near the promoter regions, which also allows us to determine the primary isoforms expressed in the MOE (Supplementary Fig. 2).

Next, we screened for the population-specific peaks using DiffBind package (version 2.8.0). By quantitatively comparing ATAC-seq peaks in TAAR and all mature OSNs, we were able to identify 6093 differential peaks with 3290 peaks enriched in OMP-positive OSNs and 2803 peaks enriched in TAAR OSNs (Fig. 2b). We then focused on two population-specific peaks in the genomic regions surrounding the *Taar* cluster. We found two peaks of about 900 bp that were highly enriched in TAAR OSNs, suggesting that they may function as putative TAAR enhancers to regulate *Taar* gene expression (Fig. 2c). We termed these sequences TAAR enhancer 1 and TAAR enhancer 2. TAAR enhancer 1 is positioned between *Taar1* and *Taar2*, while TAAR enhancer 2 is located between *Taar6* and *Taar7a*. Quantitative analysis of peak intensities after normalizing to the median values in each sample revealed a sixfold and fourfold increase for TAAR enhancer 1 and TAAR enhancer 2 in TAAR OSNs compared to OMP-positive OSNs, respectively (Fig. 2d). Thus far, 63 OR enhancers—known as "Greek islands"—have been identified and shown to contribute to *OR* gene choice in OR OSNs[15,16]. We therefore tested whether TAAR OSNs have limited chromatin accessibility at OR enhancers. Unexpectedly, we observed similar chromatin accessibility in TAAR OSNs to the whole mature OSN population (Fig. 2e). The normalized ATAC-seq peak intensities at all of the OR enhancers showed positive linear correlation with the Pearson correlation coefficient of 0.89 ($p < 0.0001$, Fig. 2f). The co-existence of putative TAAR enhancers and OR enhancers in TAAR OSNs suggests a possibility that they may form an enhancer hub to facilitate *Taar* gene choice. Consistent with this notion, we observed dramatic reduction of *Taar* gene expression after deletion of Lhx2 or Ldb1, which were shown to facilitate the formation of the OR enhancer hub[17] (Supplementary Fig. 3a). Furthermore, we successfully identified conserved motifs, including 2 Lhx2-binding motifs, 1 Lhx2/Ebf co-bound composite motif in TAAR enhancer 1, and 2 Lhx2-binding motifs, 1 Ebf-binding motif, and 1 Lhx2/Ebf composite motif in TAAR enhancer 2, similar to those identified in OR enhancers[16,17] (Supplementary Fig. 3b, c). Ldb1 could be further recruited to facilitate OR enhancer hub formation in an Lhx2-dependent manner[17]. This implies that the Lhx2/Ebf transcriptional activities as well as enhancer hub formation by Ldb1 could both contribute to functional activation of the two putative TAAR enhancers. However, the Hi-C analysis is still required to provide the direct evidence for the enhancer hub formation in TAAR OSNs in the future study.

**Evolutionary conservation of the two putative TAAR enhancers.** The *Taar* gene family is hypothesized to emerge after the segregation of jawed from jawless fish[23]. To identify the evolutionary origin of the two putative TAAR enhancers, we compared the *Taar* cluster sequences from 8 Glires, 21 Euarchontoglires, 40 placental mammals, and 60 vertebrates (Supplementary Fig. 4a). The two putative TAAR enhancers are among the most conserved

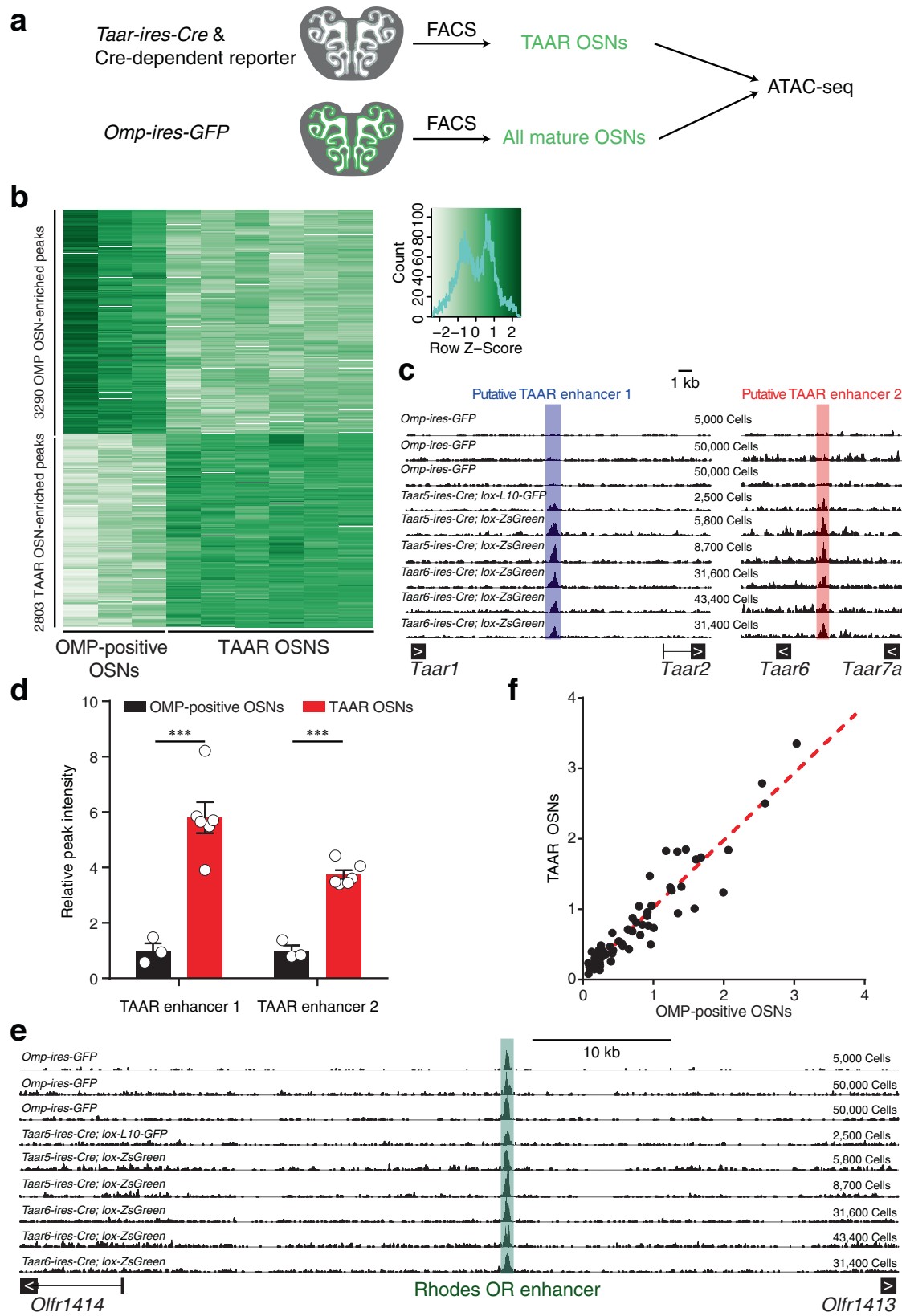

regions in addition to *Taar* genes (Fig. 3a). We then focused on the two TAAR enhancers and searched for the publicly available genome databases with the mouse sequences of the TAAR enhancers. We were able to retrieve homologous sequences from Eutheria or placental mammals but failed to find any homologous sequences from closely related Metatheria or marsupial

mammals. We next selected one representative species for each Eutheria order, including human, chimpanzee, tarsier, rat, rabbit, hedgehog, pig, cow, sperm whale, horse, cat, big brown bat, ele-phant, and armadillo. We selected koala and opossum as repre-sentative species of Metatheria and platypus as a representative species of Prototheria (Fig. 3b). The nucleotide sequences of

**Fig. 2 Identification of two putative TAAR enhancers. a** Schematic illustration of the strategy to identify putative TAAR enhancers using ATAC-seq. **b** 6093 differential peaks with 3290 peaks enriched in OMP-positive OSNs and 2803 peaks enriched in TAAR OSNs were identified with criterion of $q <$ 0.05. **c** ATAC-seq signals across the *Taar* cluster showed two putative TAAR enhancers that were enriched in TAAR OSNs. ATAC-seq signals were normalized to median peak values of each sample. Below the signal tracks, exons of the *Taar* genes were depicted. Arrows inside exons indicate direction of sense strand. Shaded regions indicate ATAC-seq peaks called by MACS2. **d** ATAC-seq peak intensities of two TAAR enhancers in OMP-positive OSNs and TAAR OSNs were normalized to median peak values and plotted. **e** ATAC-seq signals of Rhodes, an OR enhancer, showed similar chromatin accessibility in OMP-positive OSNs and TAAR OSNs. ATAC-seq signals were normalized to median peak values of each sample. Below the signal tracks, exons of two surrounding *OR* genes were depicted. Arrows inside exons indicate direction of sense strand. Shaded regions indicate ATAC-seq peaks called by MACS2. **f** Correlation between normalized ATAC-seq peak intensities of all 63 OR enhancers in TAAR OSNs (*Y*-axis) and those in OMP-positive OSNs (*X*-axis). Pearson correlation coefficient $r = 0.89$ (two-sided, no adjustments were made), $p < 0.0001$. In **d**, ***$q < 0.001$, by the Wald test (two-sided). In **d**, **f**, $n = 3$ for OMP-positive OSNs, $n = 6$ for TAAR OSNs. In **d**, data are presented as mean values ± SEM.

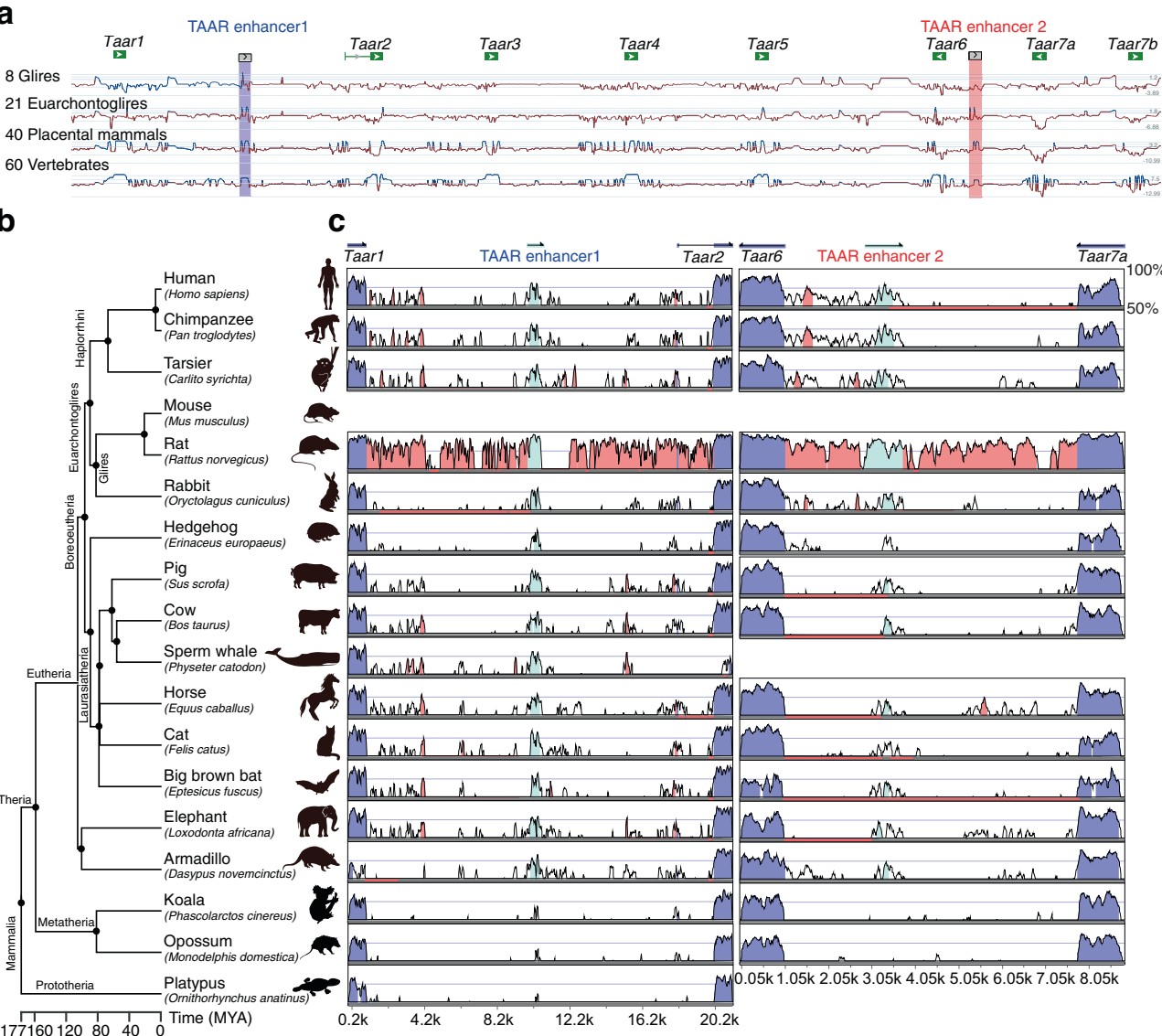

**Fig. 3 Evolutionary conservation of the two TAAR enhancers. a** Conservation of the *Taar* cluster ranging from *Taar1* to *Taar7b* in mouse with that in 8 Glires, 21 Euarchontoglires, 40 placental mammals, and 60 vertebrates using phyloP methods. Sites predicted to be conserved were assigned positive scores and shown in blue. Peaks represent the conserved regions. **b** Phylogenetic tree of representative species selected from Eutheria, Metatheria, and Prototheria order. Each branch length was calculated from divergence times obtained from TimeTree. MYA million years ago. **c** Nucleotide percent identity plots of TAAR enhancer 1 (left) and TAAR enhancer 2 (right) sequences from different mammals compared to corresponding mouse sequences using VISTA. The arrows indicate the transcriptional orientations of *Taar* genes in mouse genome. Conservation between 50 and 100% are shown as peaks. Highly conserved regions (>100 bp width and >70% identity) in exons, intergenic regions, and TAAR enhancers are shown in blue, red, and cyan, respectively.

TAAR enhancers as well as neighboring *Taar* genes from the various species were extracted and aligned with homologous mouse sequences. The VISTA plot revealed highly conserved synteny of the *Taar* genes in all of the selected species (Fig. 3c). In contrast, the two TAAR enhancers are only conserved in placental mammals but not in marsupial mammals (Fig. 3c). Interestingly, TAAR enhancer 1 is present in all placental mammals except very few species, such as dolphin and gibbon (Supplementary Fig. 4b), while TAAR enhancer 2 is less conserved and more variable in different species of placental mammals. We observed loss of TAAR enhancer 2 in dog, sea lion, and seal of the Carnivora order and in whale and dolphin of the Cetartiodactyla order, and also in gibbon. In addition, we found duplications of TAAR enhancer 2 in cow and sheep of the Cetartiodactyla order and also in horse of the Perissodactyla order, which may be accompanied by genome duplications of *Taar6* and *Taar7* family members (Supplementary Fig. 4c). In conclusion, the sequences of the two TAAR enhancers are conserved in placental mammals, supporting the notion that they play an essential role in *Taar* gene regulation.

**Deletion of TAAR enhancer 1 results in specific reduction of TAAR OSNs**. To examine the function of TAAR enhancer 1, we generated TAAR enhancer 1 knockout mice using CRISPR-Cas9 genome editing system (Fig. 4a, b). We then performed RNA-seq on the MOE dissected from homozygous, heterozygous, and wild-type littermates to obtain transcriptomic profiles for each genotype. We first compared the gene expression levels in TAAR enhancer 1 homozygous knockout mice and wild-type mice. Using the criteria of $q < 0.05$ ($q$ value is false discovery rate-corrected $p$ value) and fold change >1.5-fold, we detected 14 differentially expressed genes (DEGs), 9 of which were *Taar* genes (Supplementary Fig. 5a). We then carefully inspected the expression changes of all *Taar* genes and genes close to the *Taar* cluster (Fig. 4c and Supplementary Fig. 5b). Of the 14 functional olfactory *Taar* genes (*Taar1* is not expressed in the MOE and *Taar7c* is a pseudogene), the mRNA levels of 8 receptors (*Taar2, Taar3, Taar4, Taar5, Taar6, Taar7a, Taar7b,* and *Taar9*) were significantly decreased in homozygous TAAR enhancer 1 knockout mice (fold change <−1.5, $q < 0.05$), whereas the mRNA level of a single *Taar* gene, *Taar7e*, was significantly increased (fold change >1.5, $q < 0.05$). In addition, the mRNA levels of 3 *Taar* genes (*Taar8a, Taar8b,* and *Taar8c*) tended to decrease (fold change <−1.5, $q > 0.05$), and those of 2 *Taar* genes (*Taar7d* and *Taar7f*) were unaltered (−1.5 < fold change <1.5, $q > 0.05$). It is intriguing that deletion of TAAR enhancer 1 had dramatic effects on expression of *Taar* genes that are located in both proximal and distal positions within the *Taar* cluster but had minimal effects on the expression of *Taar* genes in the center of the cluster. This observation suggests that TAAR enhancer 1 might form a loop to affect *Taar* gene expression over around 180 kb genomic region. The effect of TAAR enhancer 1 knockout is specific to *Taar* genes, as the expression levels of other genes located next to the *Taar* cluster were not changed (Fig. 4c). We also checked the expression patterns of *OR* genes and found no significant changes for all of the *OR* genes ($q > 0.05$), further suggesting the specific effect of TAAR enhancer 1 on *Taar* genes (Fig. 4d, Supplementary Fig. 5c, and Supplementary Data 1).

**TAAR enhancer 1 regulates the probability of *Taar* gene choice**. The observed changes in *Taar* mRNA expression may be due to altered probability of *Taar* gene choice or altered *Taar* transcript levels. To distinguish between these two possibilities, we performed in situ hybridization experiments and quantified cells with positive mRNA expression signals (Fig. 4e, f).

Consistent with our RNA-seq results, cells expressing *Taar2, Taar3,* and *Taar5* were totally abolished in homozygous TAAR enhancer 1 knockout mice, while the numbers of cells expressing *Taar4, Taar6, Taar7a, Taar7b, Taar7d, Taar8s,* and *Taar9* were significantly decreased. And the numbers of cells expressing *Taar7e* and *Taar7f* were not significantly changed (Fig. 4f and Supplementary Table 2). Furthermore, we found that the changes in the numbers of TAAR OSNs were in positive linear correlation with changes in mRNA expression from the RNA-seq data (Pearson correlation coefficient r = 0.94, $p = 4.9 \times 10^{-6}$, Fig. 4g and Supplementary Table 3). Thus, like the H, P, and J elements, TAAR enhancer 1 regulates the probability of receptor gene choice rather than the transcript levels per cell[19,22]. In agreement with the RNA-seq data, we did not detect any significant differences in the number of cells expressing two *OR* genes (dorsally expressed class I *Olfr578* and ventrally expressed class II *Olfr1507*) between homozygous and wild-type TAAR enhancer 1 knockout mice (Fig. 4e, f).

To further validate these findings, we performed immunohistochemistry analysis on the MOE using specific antibodies against TAAR5 and TAAR6 proteins. Again, we observed complete abolishment of TAAR5-positive OSNs (Fig. 4h) and decreased number of TAAR6-positive OSNs in homozygous TAAR enhancer 1 knockout mice (Supplementary Fig. 5d), whereas the numbers of OSNs expressing Olfr552 (class I OR) and Olfr1507 were not significantly changed (Fig. 4h and Supplementary Fig. 5d).

The dramatic reduction of TAAR OSNs in TAAR enhancer 1 knockout mice may be due to neuronal cell death as a result of the inability to express functional TAARs. Therefore, we counted the number of apoptotic cells in the MOE by caspase-3 staining. We did not find significant difference between homozygous and wild-type TAAR enhancer 1 knockout mice (Supplementary Fig. 5e), suggesting that deletion of TAAR enhancer 1 does not induce cell death. It is possible that OSNs originally choosing *Taar*s arrest neuronal differentiation after TAAR enhancer 1 deletion.

**Deletion of TAAR enhancer 2 causes decrease of TAAR OSNs in a pattern that is slightly different from deletion of TAAR enhancer 1**. Next, to examine the function of TAAR enhancer 2, we generated TAAR enhancer 2 knockout mice by CRISPR-Cas9 genome editing (Fig. 5a, b). RNA-seq on the MOE dissected from homozygous, heterozygous, and wild-type littermates were performed to obtain transcriptomic profiles for each genotype. Using the criteria of $q < 0.05$ and fold change >1.5-fold, we detected 6 DEGs, 5 of which were *Taar* genes (Supplementary Fig. 6a). Of the 14 functional *Taar* genes, the mRNA levels of 5 receptors (*Taar3, Taar5, Taar6, Taar7a,* and *Taar9*) were significantly decreased in homozygous TAAR enhancer 2 knockout mice compared to their wild-type littermates (fold change <−1.5, $q < 0.05$). In addition, the mRNA levels of 5 *Taar* genes (*Taar4, Taar7f, Taar8a, Taar8b,* and *Taar8c*) tended to decrease (fold change <−1.5, $q > 0.05$) and those of 4 *Taar* genes (*Taar2, Taar7b, Taar7d,* and *Taar7e*) were unaltered (−1.5 <fold change <1.5, $q > 0.05$) (Fig. 5c and Supplementary Fig. 6a, b). Similar to TAAR enhancer 1, deletion of TAAR enhancer 2 did not change the expression levels of neighbor genes in the *Taar* gene cluster or any of the *OR* genes (Fig. 5c, d, Supplementary Fig. 6c, and Supplementary Data 1). We then performed in situ hybridization experiments to quantify positive cells expressing olfactory receptor mRNAs. In accordance with our RNA-seq results, the numbers of TAAR OSNs expressing *Taar3* and *Taar5* were abolished and those expressing *Taar4, Taar6, Taar7a, Taar7f, Taar8s,* and *Taar9* were significantly decreased in homozygous TAAR enhancer 2 knockout mice (Fig. 5e, f and Supplementary

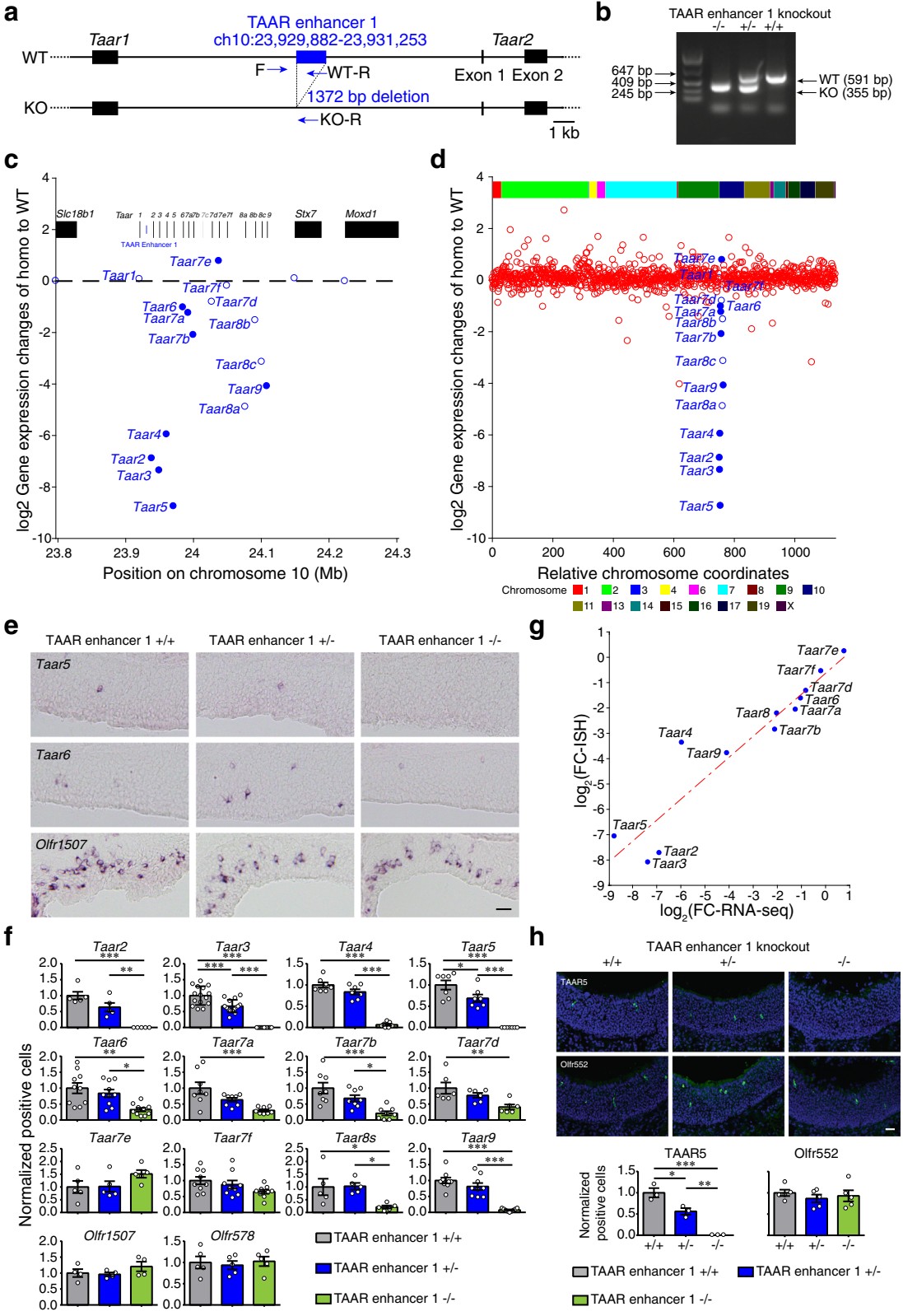

Table 4). The changes in the numbers of TAAR OSNs are in positive linear correlation with changes in mRNA expression from RNA-seq data (Pearson correlation coefficient $r = 0.88$, $p = 0.0001$), suggesting that TAAR enhancer 2 also regulates the probability of receptor gene choice (Supplementary Fig. 6d and Supplementary Table 5). As a control, we did not detect significant changes of cells expressing *Olfr578* and *Olfr1507* between

homozygous and wild-type TAAR enhancer 2 knockout mice (Fig. 5e, f). Comparison between the phenotypes of TAAR enhancer 1 and TAAR enhancer 2 knockout mice showed that cells expressing *Taar3* and *Taar5* were abolished in both knockout mice. However, we also observed subpopulations of TAAR OSNs in which the effects in TAAR enhancer 2 knockout mice were different from those in TAAR enhancer 1 knockout

**Fig. 4 TAAR enhancer 1 deletion results in massive reduction of *Taar* gene expression. a** Schematic illustration of TAAR enhancer 1 deletion using the CRISPR-Cas9 genome editing system. Arrows indicate primers used for genotype determination. WT wild type, KO knockout. **b** Representative PCR genotyping results to distinguish between wild type (+/+), heterozygous (+/−), and homozygous (−/−) TAAR enhancer 1 knockout mice using primers indicated in **a** (n ≥ 3). **c** RNA-seq analysis showed specific decreased expression of the *Taar* genes in homozygous TAAR enhancer 1 knockout mice. The log2-fold change values for the *Taar* genes and 3 surrounding genes were plotted in the *Taar* cluster region. Differentially expressed genes with criteria of $q < 0.05$ were represented by filled circles (n = 3 mice). **d** The log2-fold change values for the *Taar* genes and all of the functional *OR* genes were plotted according to their relative positions along the chromosome (*Taar* genes, blue; *OR* genes, red). Differentially expressed genes with criteria of $q < 0.05$ were represented by filled circles (n = 3 mice). **e** Representative images of *Taar5*, *Taar6*, and *Olfr1507* expression in wild type (+/+), heterozygous (+/−), and homozygous (−/−) TAAR enhancer 1 knockout mice using single color in situ hybridization. Scale bar = 25 μm. **f** The numbers of OSNs expressing *Taar* genes and 2 *OR* genes (*Olfr1507* and *Olfr578*) were quantified, and percentage of positive cell numbers in heterozygous (+/−) or homozygous (−/−) mice compared to wild type (+/+) was plotted. Data are presented as mean values ± SEM. The numbers of "n" are shown in Supplementary Table 2. **g** Correlation between log2-fold change values of positive cell numbers by in situ hybridization (Y-axis, n ≥ 4) and gene expression by RNA-seq (X-axis, n = 3) for the *Taar* genes. Pearson correlation coefficient $r = 0.94$ (two-sided, no adjustments were made), $p = 4.9 \times 10^{-6}$. **h** Top, representative confocal images of immunohistochemistry staining for TAAR5 and Olfr552 in wild type (+/+), heterozygous (+/−), and homozygous (−/−) TAAR enhancer 1 knockout mice. Blue fluorescence represented DAPI counterstaining. Scale bar = 25 μm. Bottom, percentage of positive cell numbers in heterozygous (+/−) or homozygous (−/−) mice compared to wild type (+/+) was plotted. Data are presented as mean values ± SEM (n = 3 for TAAR5 staining, n = 5 for Olfr552 staining). In **f**, **h**, *$p < 0.05$, **$p < 0.01$, ***$p < 0.001$, by one-way ANOVA and post hoc Tukey's test.

mice. For instance, the cells expressing *Taar2* were abolished in TAAR enhancer 1 knockout mice (Fig. 4e, f), but their numbers were unaltered in TAAR enhancer 2 knockout mice (Fig. 5e, f). And the numbers of cells expressing *Taar7b* and *Taar7d* were significantly decreased in TAAR enhancer 1 knockout mice (Fig. 4e, f) but were unaltered in TAAR enhancer 2 knockout mice (Fig. 5e, f). On the contrary, the numbers of cells expressing *Taar7f* were significantly decreased in TAAR enhancer 2 knockout mice (Fig. 5e, f) but were unaltered in TAAR enhancer 1 knockout mice (Fig. 4e, f). These results suggest that TAAR enhancers 1 and 2 may function jointly, redundantly, or separately to regulate the expression of different *Taar* genes.

We next verified the above observation by immunohistochemistry using TAAR5 or Olfr552 antibodies. We observed complete abolishment of TAAR5-positive OSNs, whereas OSNs expressing Olfr552 were not significantly changed (Supplementary Fig. 6e). We also counted the numbers of apoptotic cells in the MOE by caspase-3 staining and did not find significant difference between homozygous and wild-type TAAR enhancer 2 knockout mice (Supplementary Fig. 6f). This suggests that deletion of TAAR enhancer 2 may lead to neuronal differentiation arrest instead of neuronal apoptosis, analogous to the effects of TAAR enhancer 1 deletion.

**Deletion of the two TAAR enhancers results in complete elimination of TAAR OSNs.** Our results show that deletion of either one of the two enhancers decreases the expression levels of different subgroups of *Taar* genes, suggesting that they may function coordinately to regulate *Taar* gene expression. To investigate whether the whole *Taar* gene repertoire is fully controlled by the two TAAR enhancers, we generated the TAAR enhancer 1 & 2 double knockout mouse line with both enhancers deleted (Fig. 6a, b).

RNA-seq on the MOE dissected from homozygous, heterozygous, and wild-type TAAR enhancer 1 & 2 double knockout mice were performed to investigate transcriptomic profiles. Using the criteria of $q < 0.05$ and fold change >1.5-fold, we detected 13 DEGs, 12 of which were *Taar* genes (Supplementary Fig. 7a). Of the 14 functional *Taar* genes, the mRNA levels of 12 receptors (*Taar2*, *Taar3*, *Taar4*, *Taar5*, *Taar6*, *Taar7a*, *Taar7b*, *Taar7d*, *Taar7e*, *Taar7f*, *Taar8b*, and *Taar9*) were significantly decreased in homozygous TAAR enhancer 1 & 2 double knockout mice compared to their wild-type littermates (fold change <−1.5, $q < 0.05$). The mRNA levels of 2 *Taar* genes (*Taar8a* and *Taar8c*) tended to decrease (fold change <−1.5, $q > 0.05$; Fig. 6c and Supplementary Fig. 7a, b). Actually very few reads were detected

for *Taar* genes in RNA-seq data of homozygous TAAR enhancer 1 & 2 double knockout mice, and most of the *Taar* genes showed no counts at all (Supplementary Data 1). In contrast, deletion of both TAAR enhancers did not change the expression levels of neighbor genes in the *Taar* gene cluster or any of the *OR* genes (Fig. 6c, d, Supplementary Fig. 7c, and Supplementary Data 1). We then counted the numbers of TAAR OSNs in homozygous TAAR enhancer 1 & 2 double knockout mice and their littermate controls. Consistent with the RNA-seq data, all of the TAAR OSNs were completely eliminated after both of the enhancers were deleted (Fig. 6e, f and Supplementary Tables 6 and 7). Again we did not detect significant changes of cells expressing *Olfr578* and *Olfr1507* between homozygous and wild-type TAAR enhancer 1 & 2 double knockout mice (Fig. 6e, f).

We then counted the numbers of apoptotic cells in the MOE by caspase-3 staining and did not find significant difference between homozygous, heterozygous, and wild-type TAAR enhancer 1 & 2 double knockout mice (Supplementary Fig. 7d). This again suggests that deletion of the two TAAR enhancers leads to neuronal differentiation arrest instead of neuronal apoptosis.

Taken together, the above loss-of-function experiments suggest that the two TAAR enhancers we identified are specifically and absolutely required for expression of the entire *Taar* gene family members.

**TAAR enhancers are sufficient to drive reporter expression in the TAAR OSNs.** To provide further evidence for the function of the two TAAR enhancers, we then tested whether they are sufficient to drive the expression of adjacent genes in OSNs. We first detected activities of the two TAAR enhancers in zebrafish as the majority of the OR enhancers can drive reporter expression in zebrafish OSNs[15]. Indeed, the two TAAR enhancers induced GFP reporter expression in the MOE of zebrafish, similar to the OR enhancers (Supplementary Fig. 8a–c).

To further examine their functional roles, we constructed PiggyBac transgenic plasmids with one of the two TAAR enhancers placed upstream of the minimal promoter sequence from mouse Hsp68 (heat shock protein 68 kDa) and followed by the *GFP* or *tdTomato* reporter genes (Fig. 7a). We then generated transgenic mice by injecting PiggyBac plasmids into the pronucleus of fertilized eggs. For TAAR enhancer 1-GFP transgenic mice, we obtained 12 founder lines, 9 of which exhibited robust GFP expression in OSNs (Fig. 7a). We kept one TAAR enhancer 1-GFP transgenic line for breeding and further analyses. The GFP-positive OSNs were located in both dorsal and ventral domain of the MOE (Supplementary Fig. 8d), where *Taar*

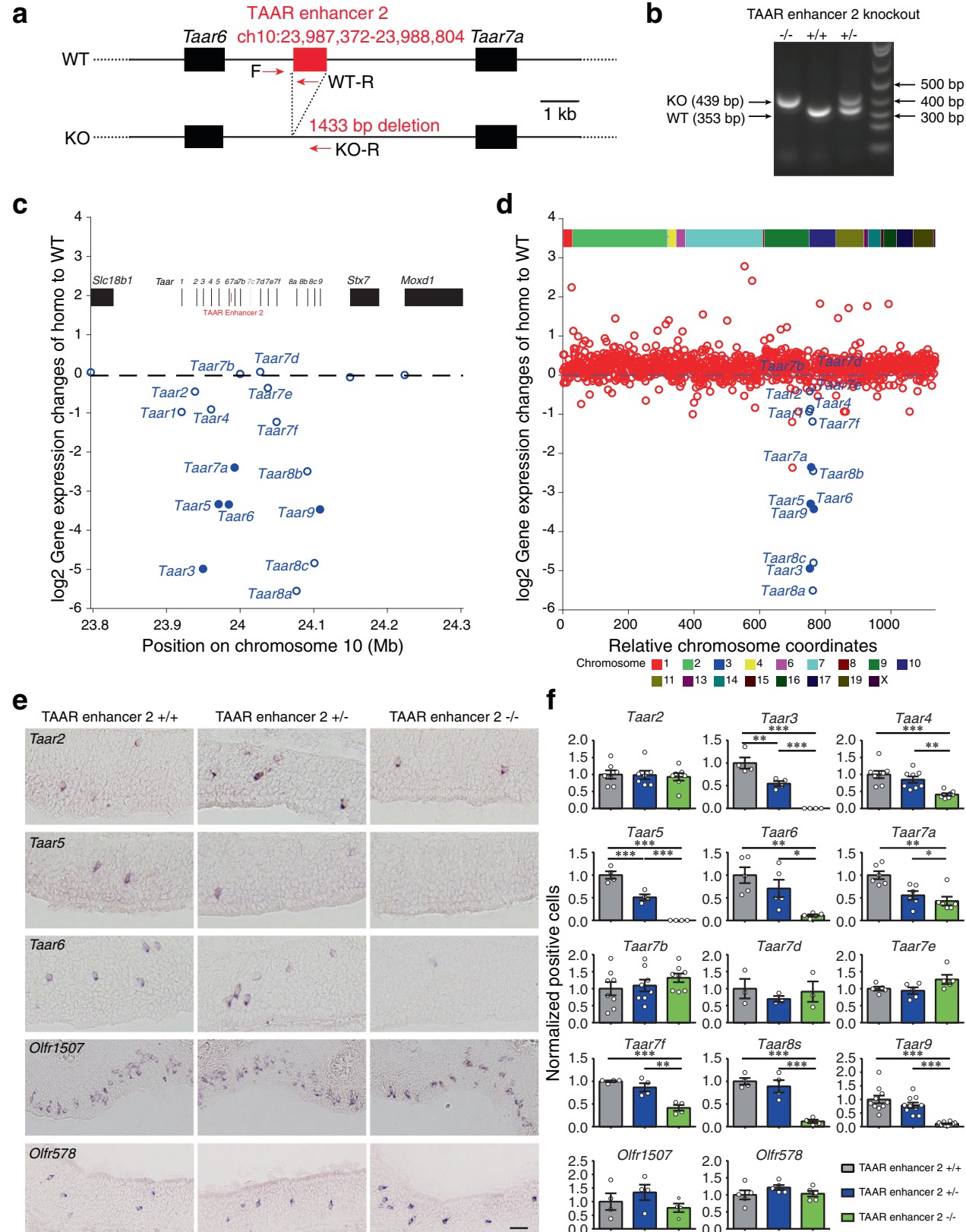

genes are normally expressed[2,25,26]. To examine whether TAAR enhancer 1 is indeed specific for TAAR OSNs, we performed RNA-seq analysis of GFP-positive and GFP-negative cells obtained by FACS sorting. We indeed observed a dramatic increase in *Taar* gene expression and decrease in *OR* gene expression in sorted GFP-positive cells compared to GFP-negative cells (Supplementary Fig. 9a). Further examination showed that all of the 14 olfactory *Taar* genes were detected at higher expression levels in GFP-positive cells than in GFP-negative cells (Supplementary Fig. 9b). Next, we analyzed whether GFP-positive OSNs overlapped with TAAR or OR OSNs using mixed *Taar* or *OR* probes (Supplementary Table 8). About 84.3%

**Fig. 5 TAAR enhancer 2 deletion also results in dramatic reduction of *Taar* gene expression. a** Schematic illustration of TAAR enhancer 2 deletion using the CRISPR-Cas9 genome editing system. Arrows indicate primers used for genotype determination. WT wild type, KO knockout. **b** Representative PCR genotyping results to distinguish between wild type (+/+), heterozygous (+/−), and homozygous (−/−) TAAR enhancer 2 knockout mice using primers indicated in **a** ($n \geq 3$). **c** RNA-seq data showed specific decreased *Taar* gene expression in homozygous TAAR enhancer 2 knockout mice. The log2-fold change values for the *Taar* genes and 3 surrounding genes were plotted in the *Taar* cluster region. Differentially expressed genes with criteria of $q < 0.05$ are represented by filled circles ($n = 3$ mice). **d** The log2-fold change values for the *Taar* genes and all of the functional *OR* genes were plotted according to their relative positions along the chromosome (*Taar* genes, blue; *OR* genes, red). Differentially expressed genes with criteria of $q < 0.05$ are represented by filled circles ($n = 3$ mice). **e** Representative images of *Taar2, Taar5, Taar6, Olfr1507,* and *Olfr578* expression in wild type (+/+), heterozygous (+/−), and homozygous (−/−) TAAR enhancer 2 knockout mice using single color in situ hybridization. Scale bar = 25 μm. **f** The numbers of OSNs expressing *Taar* genes and 2 *OR* genes (*Olfr1507* and *Olfr578*) were quantified and the percentage of positive cell numbers in heterozygous (+/−) or homozygous (−/−) mice compared to wild type (+/+) was plotted. Data are presented as mean values ± SEM. The numbers of "$n$" are shown in Supplementary Table 4. In **f**, *$p < 0.05$, **$p < 0.01$, ***$p < 0.001$, by one-way ANOVA and post hoc Tukey's test.

of GFP-positive OSNs were co-labeled with *Taar* genes, and the coexpression rate dropped to 4.1% for *OR* genes when the same number of mixed receptor probes were used. The mixed *OR* probes were composed of probes recognizing 2 class I *OR* and 6 class II *OR* genes. They may also cross-hybridize with 6 other *OR* genes (Supplementary Table 8). In addition, we prepared the mixed class I *OR* probes containing 8 class I *OR* probes (Supplementary Table 8). The coexpression rate dropped to 0.4% for mixed class I *OR* probes. We also observed a decreased coexpression rate of 0.3% for *OR* degenerate probe (Fig. 7b, c). By contrast, around 14.4% of TAAR OSNs expressed GFP reporter, and the coexpression rate dropped to 1.9% for mixed *OR* probes, 0.2% for mixed class I *OR* probes, and 0.5% for *OR* degenerate probe (Fig. 7b, c). To further reveal the identity of GFP-positive cells, we examined the coexpression rates of GFP with TAAR OSNs using individual *Taar* probes. The coexpression rates varied from 1.6 to 12.2% for different *Taar* genes, and the summed coexpression rate was 88.7% that is close to that obtained with mixed *Taar* probes (Supplementary Fig. 9c).

For TAAR enhancer 2-tdTomato transgenic mice, we obtained five founder lines, three of which exhibited robust tdTomato expression in OSNs (Fig. 7a). We kept one TAAR enhancer 2 transgenic line and examined whether TAAR enhancer 2 is indeed specific for TAAR OSNs. We performed RNA-seq analysis of tdTomato-positive and tdTomato-negative cells and also observed a dramatic increase in *Taar* gene expression and decrease in *OR* gene expression in sorted tdTomato-positive cells compared to tdTomato-negative cells (Supplementary Fig. 9a). All of the 14 olfactory *Taar* genes were detected at much higher expression levels in tdTomato-positive cells than in tdTomato-negative cells (Supplementary Fig. 9b). Coexpression analyses showed that about 73.2% of tdTomato-positive OSNs were co-labeled with *Taar* genes, and the coexpression rates dropped to 1% for mixed *OR* probes, 1.4% for mixed class I *OR* probes, and 0.3% for *OR* degenerate probe (Fig. 7d, e). On the other hand, around 13.9% of TAAR OSNs expressed tdTomato reporter, and the coexpression rates dropped to 0.2% for mixed *OR* probes, 0.7% for mixed class I *OR* probes, and 0.1% for *OR* degenerate probe (Fig. 7d, e). Further analyses using individual *Taar* probes revealed that the coexpression rates varied from 1 to 14.3% for different *Taar* genes except *Taar7a*, and the summed coexpression rate was 71.3% that is close to that obtained with mixed *Taar* probes (Supplementary Fig. 9c). Together, these results validate the specific activity of the two TAAR enhancers in TAAR OSNs.

Next, we crossed the TAAR enhancer 1-GFP transgenic mice with the TAAR enhancer 2-tdTomato transgenic mice and visualized the overlap between GFP- and tdTomato-positive cells. About 14.3% of OSNs co-expressed GFP and tdTomato driven by TAAR enhancer 1 and TAAR enhancer 2, respectively. On the other hand, 67.5 and 18.2% of OSNs expressed each reporter alone (Fig. 7f). We further analyzed the projection pattern of

OSNs into the olfactory bulb. The GFP-positive and tdTomato-positive glomeruli were largely overlapped in the dorsal region of olfactory bulb, where TAAR glomeruli cluster (called DIII domain[25]; Fig. 7g and Supplementary Fig. 8e). We also observed a few glomeruli that were positive for GFP or tdTomato alone. These data indicate that the two TAAR enhancers in transgenes are both active in some TAAR OSNs, and are separately active in other TAAR OSNs. This is similar to the function of the two endogenous TAAR enhancers that coordinately regulate *Taar* gene expression.

**TAAR enhancer 1 functions in *cis*.** Each OSN in the MOE expresses an olfactory receptor gene in a monogenic and monoallelic fashion. However, we observed similar expression levels of *Taar* genes in heterozygous TAAR enhancer 1, TAAR enhancer 2, or TAAR enhancer 1 & 2 double knockout mice compared to their wild-type littermates (Figs. 4e, f, 5e, f, and 6e, f and Supplementary Figs. 5b, 6b, and 7b), seemingly violating the monoallelic expression rule. A possible explanation for this observation is that the intact enhancers in heterozygous mice could act both in *cis* and in *trans* to regulate *Taar* gene expression in both alleles. Another explanation is that TAAR OSNs are genetically programmed to express *Taar* genes. As a result, although TAAR enhancers may just operate in *cis* like OR enhancers, TAAR OSNs can only express *Taar* genes from the allele with the functional enhancers, thereby maintaining the number of TAAR OSNs. To differentiate between these two possibilities, we crossed TAAR enhancer 1 knockout mice with *Taar2-9* cluster knockout mice, in which all of the olfactory *Taar* genes from *Taar2* to *Taar9* are deleted[43]. The successful deletion of the olfactory *Taar* genes was verified by the absence of the mRNA and protein expression of TAARs in the MOE by RNA-seq, RNA in situ hybridization, and immunohistochemistry experiments (Supplementary Fig. 10a–c). Although TAAR enhancer 2 is deleted in the *Taar2-9* cluster knockout mouse line, TAAR enhancer 1 is preserved. By crossing heterozygous TAAR enhancer 1 knockout mice (designated as ΔTAAR-enhancer1/TAAR-enhancer1 here for clarity) with heterozygous *Taar2-9* cluster knockout mice (Δ*Taar2-9*/*Taar2-9*), we acquired four genotypes of mice for further analysis: (1) genotype 1: wild-type littermates with both alleles intact, (2) genotype 2: one allele lacking TAAR enhancer 1 and the other allele intact (same genotype as heterozygous TAAR enhancer 1 knockout), (3) genotype 3: one allele lacking *Taar2-9* cluster and the other allele intact (same genotype as heterozygous *Taar2-9* cluster knockout), and (4) genotype 4: one allele lacking TAAR enhancer 1 and the other allele lacking *Taar2-9* cluster (Fig. 8a). Next, we performed RNA in situ hybridization experiments to examine the expression of *Taar* genes that were entirely eliminated (*Taar2, Taar3,* and *Taar5*) or significantly reduced (*Taar6*) in homozygous TAAR enhancer 1 knockout

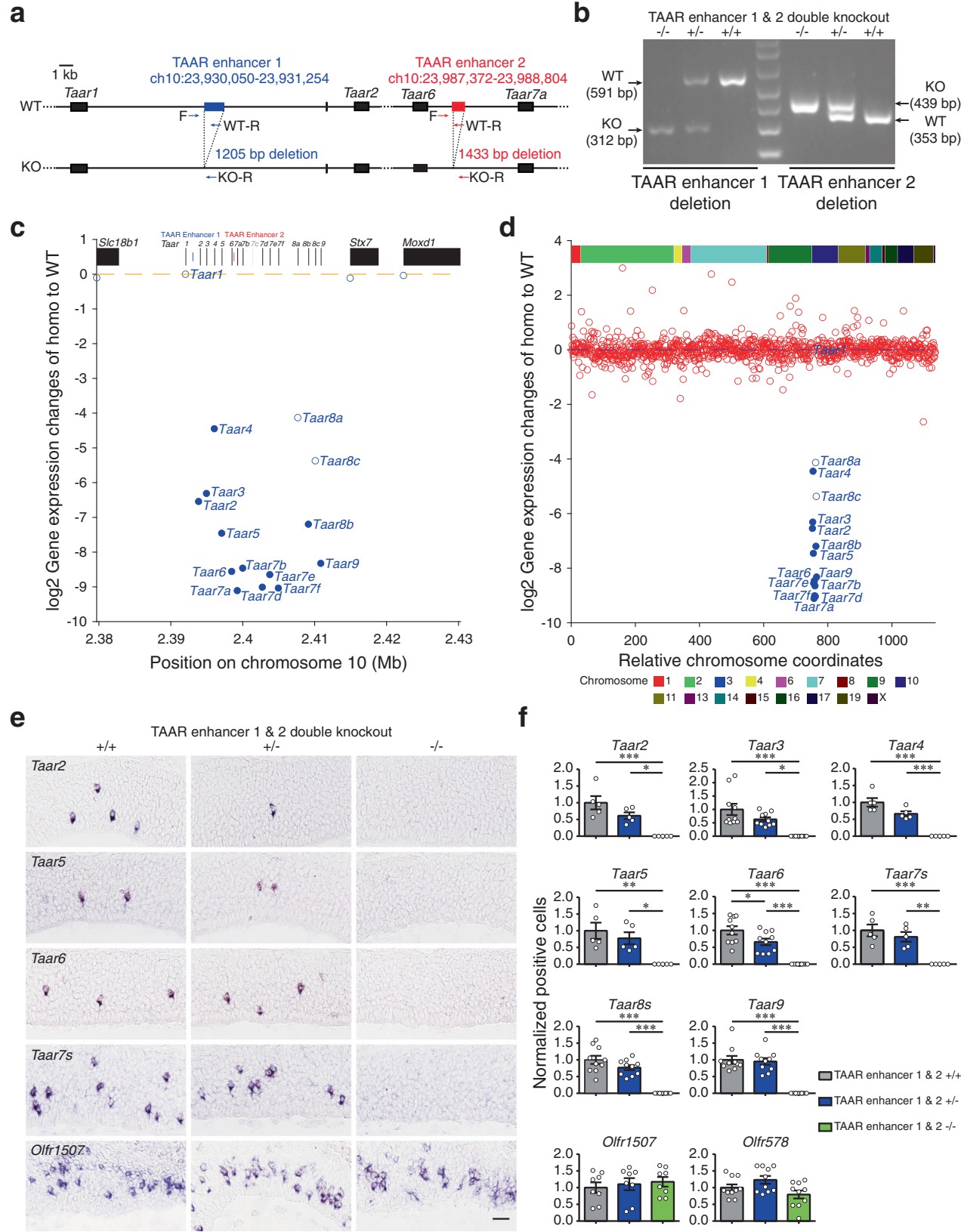

mice (Fig. 4e, f, h). In genotype 2 and 3 mice, cells expressing these *Taar* genes showed similar pattern to that in heterozygous TAAR enhancer 1 knockout and *Taar2-9* cluster knockout mice (Figs. 4e, f and 8b, c and Supplementary Fig. 10a, b), whereas cells expressing these *Taar* genes in genotype 4 mice showed similar reduction pattern to that in homozygous TAAR enhancer 1 knockout mice (Figs. 4e, f and 8b, c). These results suggest that TAAR enhancer 1 operates in *cis* to regulate *Taar* gene choice (Fig. 8a), in analogy to the OR enhancers[19–22].

**Fig. 6 Deletion of the two TAAR enhancers causes complete elimination of TAAR OSNs. a** Schematic illustration of TAAR enhancer 1 & 2 double knockout strategy using the CRISPR-Cas9 genome editing system. Arrows indicate primers used for genotype determination. WT wild type, KO knockout. **b** Representative PCR genotyping results to distinguish between wild type (+/+), heterozygous (+/−), and homozygous (−/−) TAAR enhancer 1 & 2 double knockout mice using primers indicated in **a** ($n \geq 3$). **c** RNA-seq data showed specific decreased *Taar* gene expression in homozygous TAAR enhancer 1 & 2 double knockout mice. The log2-fold change values for the *Taar* genes and 3 surrounding genes were plotted in the *Taar* cluster region. Differentially expressed genes with criteria of $q < 0.05$ are represented by filled circles ($n = 3$ for wild-type and heterozygous littermates, $n = 4$ for homozygous littermates). **d** The log2-fold change values for the *Taar* genes and all of the functional *OR* genes were plotted according to their relative positions along the chromosome (*Taar* genes, blue; *OR* genes, red). Differentially expressed genes with criteria of $q < 0.05$ are represented by filled circles ($n = 3$ for wild-type and heterozygous littermates, $n = 4$ for homozygous littermates). **e** Representative images of *Taar2*, *Taar5*, *Taar6*, *Taar7s*, and *Olfr1507* expression in wild type (+/+), heterozygous (+/−), and homozygous (−/−) TAAR enhancer 1 & 2 double knockout mice using single color in situ hybridization. Scale bar = 25 μm. **f** The numbers of OSNs expressing *Taar* genes and 2 *OR* genes (*Olfr1507* and *Olfr578*) were quantified and the percentage of positive cell numbers in heterozygous (+/−) or homozygous (−/−) mice compared to wild type (+/+) was plotted. Data are presented as mean values ± SEM. The numbers of "*n*" are shown in Supplementary Table 6. In **f**, *$p < 0.05$, **$p < 0.01$, ***$p < 0.001$, by one-way ANOVA and post hoc Tukey's test.

## Discussion

In the present study, we identified two TAAR enhancers that specifically regulate probability of *Taar* gene choice in the TAAR olfactory subsystem. Deletion of TAAR enhancer 1 in both alleles resulted in abolishment of 3 *Taar* genes (*Taar2*, *Taar3*, and *Taar5*), decrease of 8 *Taar* genes (*Taar4*, *Taar6*, *Taar7a*, *Taar7b*, *Taar8a*, *Taar8b*, *Taar8c*, and *Taar9*), increase of 1 *Taar* gene (*Taar7e*), and no significant changes of 2 *Taar* genes (*Taar7d* and *Taar7f*), while homozygous knockout of TAAR enhancer 2 resulted in abolishment of 2 *Taar* genes (*Taar3* and *Taar5*), decrease of 8 *Taar* genes (*Taar4*, *Taar6*, *Taar7a*, *Taar7f*, *Taar8a*, *Taar8b*, *Taar8c*, and *Taar9*), and no significant changes of 4 *Taar* genes (*Taar2*, *Taar7b*, *Taar7d*, and *Taar7e*). It seems that various *Taar* genes are differentially regulated by the two TAAR enhancers. For example, expression of some *Taar* genes (*Taar3* and *Taar5*) requires the presence of both enhancers and deletion of either one completely abolishes their expression. And expression of some *Taar* genes (*Taar2*) is solely dependent on TAAR enhancer 1, while expression of the majority of *Taar* genes requires either TAAR enhancer, meaning that the two enhancers have partially overlapped function. Thus, the two TAAR enhancers work coordinately to achieve expression of the whole *Taar* gene repertoire. This conclusion is further supported by analyses of the double knockout animals lacking both of the TAAR enhancers. In the double knockout mice, all of the *Taar* genes were completely abolished, suggesting that the two enhancers are fully responsible for *Taar* gene expression and hence TAAR OSN development. The findings reported by the Bozza group (in this issue of *Nature Communications*) also show that the same two enhancers (named T elements) cooperate to regulate expression of the entire olfactory *Taars*[44]. This organization resembles the regulation of protocadherin-α family genes by the combined activity of two enhancers[45] and thus provides another great model to study gene regulation by cooperative enhancers.

Our data revealed that TAAR enhancer 1, and possibly TAAR enhancer 2, operate in *cis*, which is similar to previously identified OR enhancers, including the H, P, and J elements[19–22]. However, the phenotypes of heterozygous enhancer knockout mice are very different. In mutant mice lacking the H or P elements in one allele, the number of cells expressing the specific class II OR genes regulated by those two elements is decreased to half of that in wild-type mice[19]. In contrast, when the J element or the two TAAR enhancers are deleted in one allele, the number of cells expressing class I OR genes or *Taar* genes is nearly the same as that in wild-type mice. This is consistent with the hypothesis that OSNs have dedicated cell fates to express the various olfactory receptor gene families (class I OR genes, class II OR genes, and *Taar* genes) prior to the first receptor choice[22,25,26]. When the OR enhancers that regulate expression of class II OR genes (e.g., the

H and P elements) are deleted in one allele, OSNs can express class II OR genes from the remaining intact enhancer in the other allele and >60 functional OR enhancers in both alleles. As a result, the expression probability of distinct class II OR genes regulated by the intact enhancer is largely diluted, leading to the decrease of OSNs expressing those class II OR genes by half. By contrast, when the J element or either one or both of the two TAAR enhancers are deleted in one allele, receptor selection is restricted to the intact enhancer from the other allele. Thus, the remaining intact enhancer still plays a major role in receptor selection and fill in the OSN population. As a result, the numbers of OSNs expressing class I OR genes or *Taar* genes are similar in heterozygous and wild-type animals. However, the two TAAR enhancers could also mediate *trans* genomic interactions to form a TAAR enhancer hub to facilitate *Taar* gene transcription, in analogous to OR enhancers[15,17] (see further discussion below).

The next intriguing question is how the cell fate of OSNs is determined prior to the first receptor choice. OSNs in the MOE are continuously renewed from GBCs. It is conceivable that certain cell-specific transcription factors, nucleosome remodeling complexes, and epigenetic regulators predefine the cell fate of OSNs during stem cell differentiation. Identification of such factors will greatly advance our understanding of the development of the distinct olfactory subsystems. Besides, TAAR enhancers emerge after the separation of placental mammals from marsupial mammals and are not found in zebrafish. However, our transient reporter assay showed that the two mouse TAAR enhancers are capable of driving reporter expression in the nose of zebrafish larvae (Supplementary Fig. 8a–c). This result has two implications: (1) the teleost species may utilize other uncharacterized teleost-specific enhancers to regulate expression of the largely expanded *Taar* genes (e.g., 112 members in zebrafish); (2) although TAAR enhancers are not conserved in zebrafish, the regulatory factors proposed above might be conserved to fulfill the TAAR enhancer activities.

The *Taar* gene cluster and OR gene clusters are sequestered in different nuclear compartments. In OR OSNs, silent OR gene clusters aggregate to form ~5 heterochromatic foci in the center of cell nuclei, while the *Taar* gene cluster is localized to the thin rim at the nuclear periphery[18,40,46]. However, the nuclear locations of the *Taar* gene cluster and the OR gene clusters in TAAR OSNs have not been carefully examined. Here we found that the heterochromatic histone modifications of the *Taar* gene cluster differ between the two olfactory subsystems. Although OR gene clusters are covered by H3K9me3 heterochromatin modifications in both TAAR and OR OSNs, the same heterochromatin marks of *Taar* gene cluster are only present in TAAR OSNs. This observation indicates that the conformation of the *Taar* gene cluster might be different in the

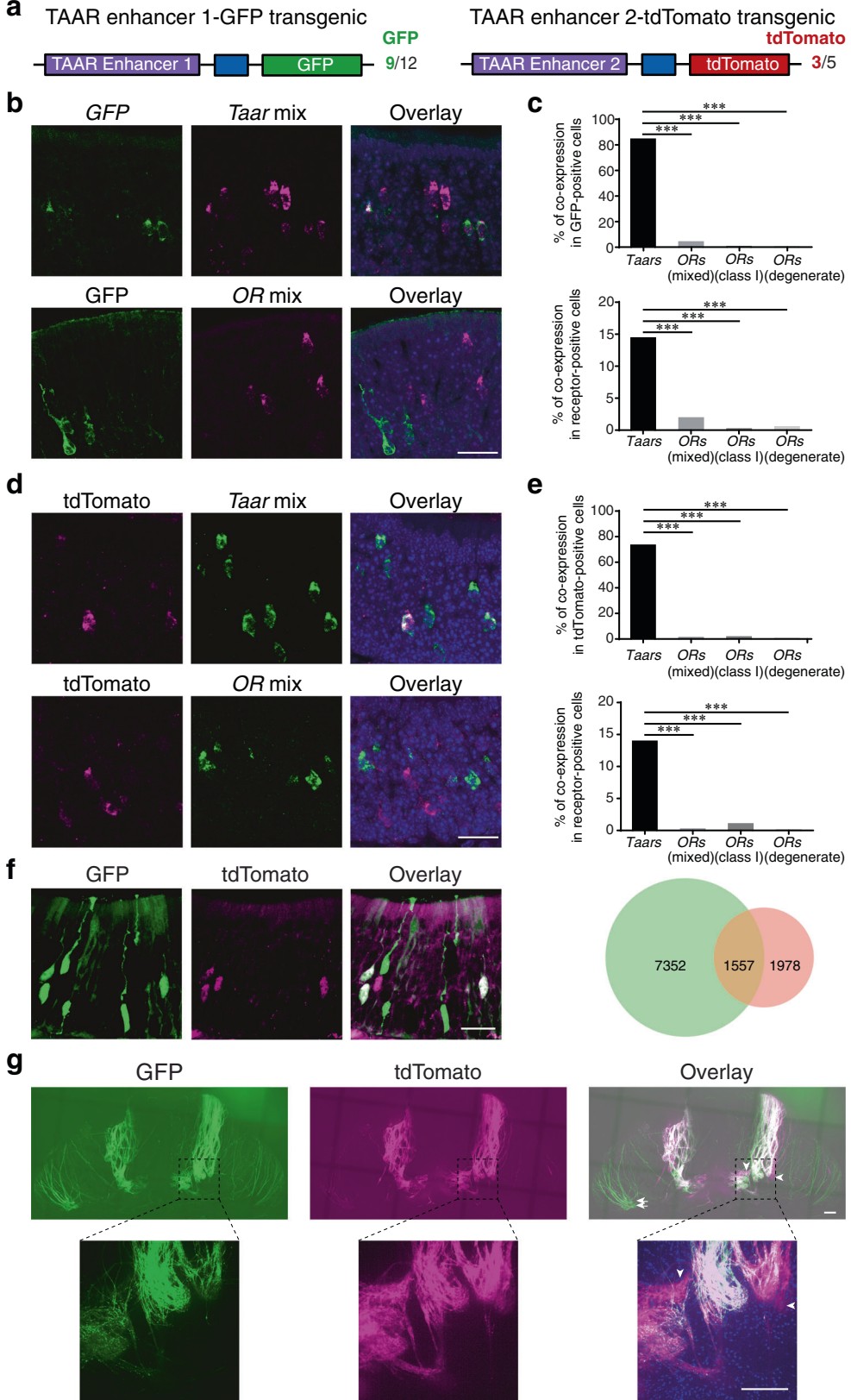

nucleus of TAAR and OR OSNs. On the other hand, activation of receptor genes is often accompanied by a shift in their nuclear localization. In OR OSNs, the active *OR* allele escapes from the central heterochromatin foci into euchromatic territory with the help of OR enhancer hub[17,46]. In TAAR OSNs, the active *Taar* allele is thought to transition from the peripheral nuclear lamina to a more permissive interior euchromatin center[40]. It is very likely that the heterochromatin formation on the *Taar* cluster contributes to the relocation of *Taar* genes from the nuclear lamina in TAAR OSNs, which is consistent with the recent finding that the levels of heterochromatin are strongly correlated with the frequency of Hi-C contacts among

**Fig. 7 TAAR enhancer 1 and enhancer 2 possess TAAR OSN-specific enhancer activity. a** Schematic illustration of the design strategy for TAAR enhancer 1-GFP (left) and TAAR enhancer 2-tdTomato (right) transgene constructs. The boxes in dark blue represent Hsp68 minimal promoter. The numbers of founders having GFP-positive and tdTomato-positive cells in the MOE were also indicated. We kept one founder for each enhancer transgenic line and used them for further experiments. **b** Top, confocal images of two-color in situ hybridization using probes for *GFP* (green) and all of the *Taar* genes (magenta) in a TAAR enhancer 1-GFP transgenic mouse. Bottom, confocal images of in situ hybridization of mixed probes of *OR* genes (magenta) and the following immunohistochemistry by the GFP antibody (green) in a TAAR enhancer 1-GFP transgenic mouse. Scale bar = 25 μm ($n \geq 3$). **c** Top, bar plots showing the percentages of GFP-positive cells that were co-labeled with mixed *Taar* probes, mixed *OR* probes, mixed class I *OR* probes, or *OR* degenerate probe (1083 out of 1284 cells for mixed *Taar* probes, 25 out of 616 cells for mixed *OR* probes, 2 out of 471 cells for mixed class I *OR* probes, 1 out of 351 cells for *OR* degenerate probe). Bottom, bar plots showing the percentages of TAAR or OR OSNs that co-expressed GFP using mixed *Taar* probes, mixed *OR* probes, mixed class I *OR* probes, or *OR* degenerate probe (564 out of 3920 cells for mixed *Taar* probes, 17 out of 898 cells for mixed *OR* probes, 2 out of 1153 cells for mixed class I *OR* probes, 1 out of 203 cells for *OR* degenerate probe). **d** Confocal images of in situ hybridization of mixed probes of *Taar* genes or *OR* genes (green) and the following immunohistochemistry by the tdTomato antibody (magenta) in a TAAR enhancer 2-tdTomato transgenic mouse. Scale bar = 25 μm ($n \geq 3$). **e** Top, bar plots showing the percentages of tdTomato-positive cells that were co-labeled with mixed *Taar* probes, mixed *OR* probes, mixed class I *OR* probes, or *OR* degenerate probe (817 out of 1116 cells for mixed *Taar* probes, 9 out of 896 cells for mixed *OR* probes, 5 out of 349 cells for mixed class I *OR* probes, 2 out of 657 cells for *OR* degenerate probe). Bottom, bar plots showing the percentages of TAAR or OR OSNs that co-expressed tdTomato using mixed *Taar* probes, mixed *OR* probes, mixed class I *OR* probes, or *OR* degenerate probe (817 out of 5863 cells for mixed *Taar* probes, 9 out of 4726 cells for mixed *OR* probes, 5 out of 748 cells for mixed class I *OR* probes, 2 out of 2713 cells for *OR* degenerate probe). **f** Representative confocal images showing cells with GFP and tdTomato signals in TAAR enhancer 1-GFP; TAAR enhancer 2-tdTomato transgenic mice. Right, the numbers of cells expressing each reporter alone (GFP, tdTomato) or together were counted. Scale bar = 25 μm ($n \geq 3$). **g** Whole-mount fluorescent images of the dorsal view of olfactory bulb in TAAR enhancer 1-GFP; TAAR enhancer 2-tdTomato transgenic mice. The area in the black dashed box is enlarged on the bottom. The white arrows indicate GFP-specific glomeruli, while the white arrow heads indicate tdTomato-specific glomeruli. Scale bar = 200 μm ($n \geq 3$). In **c**, **e**, ***$p < 0.001$ by Fisher's exact test (two-sided).

the *OR* clusters[41]. In addition, the movement of the *Taar* gene cluster away from the repressive heterochromatin environment may need assistance of the two TAAR enhancers. The two TAAR enhancers could act in *trans* to form a TAAR enhancer hub with OR enhancers based on the finding that OR enhancers are also open in TAAR OSNs. This together with the distinct heterochromatin decoration on the *Taar* cluster raise an interesting possibility that the TAAR olfactory subsystem is hijacking the pre-existed apparatus for singular *OR* gene choice by simply developing the two TAAR enhancers. Moreover, the recent development and application of single-cell chromatin conformation capture method[18,47] could help to reveal the three-dimensional structure of the TAAR enhancer hub in TAAR OSNs and the potential interaction between the two TAAR enhancers with distinct *Taar* genes at higher resolution. This may explain the differential regulation of *Taar* genes by the two TAAR enhancers. Therefore, revealing the spatial organization of the *Taar* gene cluster and the two TAAR enhancers in TAAR vs. OR OSNs would further advance our understanding of how the entire *Taar* gene repertoire is regulated.

## Methods

**Mice**. All mouse experiments were approved by the Animal Ethics Committee of Shanghai Jiao Tong University School of Medicine and the Institutional Animal Care and Use Committee (Department of Laboratory Animal Science, Shanghai Jiao Tong University School of Medicine, animal protocol number A-2016-049). Mice were housed in standard conditions with temperatures of 65–75 °F (18–23 °C), 40–60% humidity, and a 12-h light/12-h dark cycle. They have access to rodent chow and water ad libitum. Both male and female mice were used for experiments. Generation of genetically manipulated mouse lines including *Omp-ires-GFP* and *Taar2-9* cluster knockout mice have been described previously[14,43]. The genotyping primers are included in Supplementary Table 9.

We used gene targeting in embryonic stem (ES) cells to generate two strains of mice, *Taar5-ires-Cre* and *Taar6-ires-Cre*, that express Cre recombinase in TAAR5- and TAAR6-expressing OSNs. To generate the *Taar5-ires-Cre* allele, the DNA sequence encoding Cre was introduced into the 3' untranslated sequence in the *Taar5* gene immediately following the stop codon with an IRES sequence preceding the Cre sequence to allow bicistronic expression of TAAR5 and Cre[48]. To select positive ES clones, the target construct included a cassette of the neomycin gene flanked by FRT sites. The neomycin gene was removed by crossing the mice to a germline Flp recombinase line. The same strategy was used to generate the *Taar6-ires-Cre* allele.

TAAR enhancer 1 or TAAR enhancer 2 was deleted using the CRISPR-Cas9 genome editing technique with co-injection of spCas9 mRNA and single guide

RNAs (sgRNAs). The plasmids used for making spCas9 mRNA and sgRNAs are summarized in Supplementary Table 10. Two groups of sgRNAs were designed at upstream and downstream of the target sequences. The targeted sequences for TAAR enhancer 1 were 5'-TGGTTGTGAGTTGCTTGTGG-3' (sgRNA1 sense), 5'-TCAGCCTGTTAATTACCTGA-3' (sgRNA1 antisense), 5'-AGAACTTTCAGAG AGTTCCC-3' (sgRNA2 sense), 5'-GAACCCAGAACTGACTTTTG-3' (sgRNA2 antisense), 5'-TATTCTAGAAATACAGATGT-3' (sgRNA3 sense), and 5'-AGC ATCCTGGAGGTGAAATG-3' (sgRNA3 antisense). The targeted sequences for TAAR enhancer 2 were 5'-GTAAATAAAAACTTTCCCTC-3' (sgRNA1 sense), 5'-CTCCATCGTCACAAAGCCTG-3' (sgRNA1 antisense), 5'-CCCTCAAAAAG TTTGTTTTT-3' (sgRNA2 sense) and 5'-CAGGTCTTTTTTAGTGGACT-3' (sgRNA2 antisense). The RNA mixtures (50 ng spCas9 mRNA per μl and 100 ng sgRNA mixture per μl) were injected into zygotes of C57BL/6J mice. The two-cell stage embryos were then transferred into surrogate mothers. We obtained 11 candidate founders for TAAR enhancer 1 knockout mice and 8 candidate founders for TAAR enhancer 2 knockout mice. We eventually established one TAAR enhancer 1 knockout line with a 1372-bp deletion (mm10, chr10: 23,929,882–23,931,253) and one TAAR enhancer 2 knockout line with a 1433-bp deletion (mm10, chr10: 23,987,372–23,988,804) that were used for RNA-seq, in situ hybridization, and immunohistochemistry experiments.

TAAR enhancer 1 & 2 double knockout mice were generated by injecting spCas9 mRNA and the above sgRNAs targeting TAAR enhancer 1 into zygotes of TAAR enhancer 2 homozygous or heterozygous mice. We obtained 3 candidate founders and eventually established one double knockout line with a 1205-bp deletion (mm10, chr10: 23,930,050–23,931,254) for TAAR enhancer 1 and a 1433-bp deletion (mm10, chr10: 23,987,372–23,988,804) for TAAR enhancer 2.

The TAAR enhancer 1 or TAAR enhancer 2 transgenic mice were generated using the PiggyBac Transposon system. Briefly, TAAR enhancer 1 (mm10, chr10: 23,930,200–23,931,190, 990 bp) or TAAR enhancer 2 (mm10, chr10: 23,987,501–23,988,362, 862 bp) was cloned into the modified PiggyBac transposon vector, in which enhancers were followed by Hsp68 minimal promoter and GFP (for TAAR enhancer 1) or tdTomato (for TAAR enhancer 2) sequences. The plasmid information is included in Supplementary Table 10. The validated plasmid (100 ng per μl) as well as PiggyBac transposase mRNA (25 ng per μl) were co-injected into fertilized eggs of C57BL/6J mice. The two-cell stage embryos were then transferred into surrogate mothers.

**Fluorescence-activated cell sorting**. Mice were sacrificed with $CO_2$ followed by cervical dislocation. The MOE tissue was dissected and cells were dissociated using Papain Dissociation System (Worthington Biochemical) following the manufacturer's instructions with minor modifications. Briefly, dissociation reaction was incubated at 37 °C for 15 min. The tissue was triturated for 10–15 times with a cut P1000 pipette tip. Cells were then filtered by 40-μm strainer (Falcon) and centrifuged at $400 \times g$ for 2 min. Cell pellets were resuspended in Dulbecco's modified Eagle's medium (Gibco) and kept on ice for sorting. OSNs were sorted on a FACSJazz Cell Sorter (BD) or MoFlo Astrios EQ (Beckman Coulter) with a 488- or 561-nm laser. A representative set of density plots (with gates) is shown in Supplementary Fig. 1. Sorted cells were subsequently proceeded to ChIP-seq, ATAC-seq, and RNA-seq experiments.

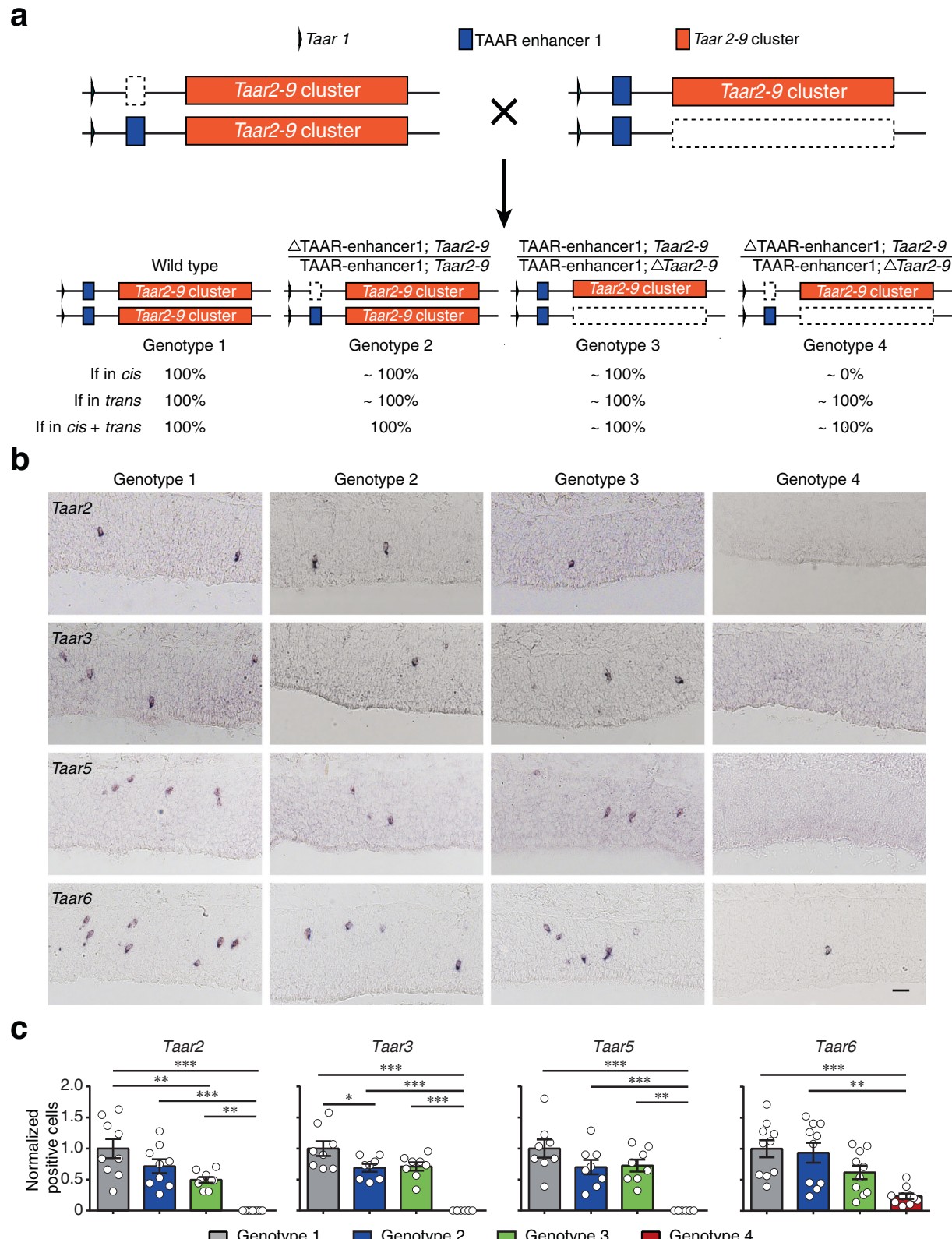

**Fig. 8 TAAR enhancer 1 operates in *cis*. a** Schematic illustration of four different mouse genotypes by crossing heterozygous TAAR enhancer 1 knockout mice with heterozygous *Taar2-9* cluster knockout mice. According to the assumption that TAAR enhancer 1 functions in *cis*, in *trans*, or in *cis* plus *trans*, the hypothetical percentage of positive cell numbers in different genotypes were indicated by normalizing to wild-type (genotype 1) mice (set as 100%). **b** Representative images of *Taar2*, *Taar3*, *Taar5*, and *Taar6* expression in four different genotypes is indicated by using a single-color in situ hybridization. Scale bar = 25 µm. **c** The numbers of OSNs expressing *Taar2*, *Taar3*, *Taar5*, and *Taar6* were quantified and the percentage of positive cell numbers normalized to wild type was plotted. Data are presented as mean values ± SEM ($n = 9$ for *Taar2* detection, $n = 8$ for *Taar3* detection, $n = 8$ for *Taar5* detection, $n = 10$ for *Taar6* detection). In **c**, $*p < 0.05$, $**p < 0.01$, $***p < 0.001$, by one-way ANOVA and post hoc Tukey's test.

**Ultra-low-input native ChIP-seq (ULI-ChIP-seq)**. ULI-ChIP-seq of H3K9me3 was performed as previously described[49] except for the library preparation portion. For each reaction, 1000–10,000 cells and 0.25 μg of H3K9me3 antibody (ABclonal, A2360) were used. Libraries were prepared with an NEBNext Ultra Kit (NEB) and 12–15 PCR cycles. Reads were mapped to the mouse reference genome (mm10) with Bowtie 2 (version 2.2.9) using the default parameters.

**ATAC-seq**. ATAC-seq was performed as previously described with minor modifications[50]. During quantitative PCR-based library quantification, a length of 300 bp was used in concentration calculation. Reads were mapped to the mouse reference genome (mm10) using Bowtie 2 (version 2.2.9) with the following parameters: --local –X 2000. Significant ATAC-seq peaks were identified using MACS2 (version 2.1.1.20160309) with the "--nomodel --shift −100 --extsize 200 --keep-dup all" parameters and the default threshold of $q$ value (adjusted $p$ value) at 0.05. Differential peak analysis between TAAR OSNs and OMP-positive mature OSNs was performed using DiffBind (version 2.8.0) in R (version 3.5.1) with standard parameters.

**RNA-seq**. Total RNAs of FACS enriched cells or the whole MOE tissues were extracted with a RNeasy Mini Kit (Qiagen) with DNase treatment. Libraries of FACS-enriched cells were prepared with the Smart-seq2 procedure[51], similar to our previous work[38]. Libraries of the whole MOE tissues were prepared with an NEBNext Ultra Directional RNA Kit (NEB) with polyA bead selection. Reads were mapped to the mouse reference genome (mm10) with Hisat2 (version 2.1.0) and quantified with Cufflinks (version 2.2.1). Raw read counts mapped to genes were calculated using featureCounts from Subread package (version 1.6.2). DEG analysis was performed using DESeq2 1.20.0 package with the default settings. Significantly changed genes were identified with $q$ value cutoff of 0.05.

**Whole-mount imaging**. Unfixed whole-mount olfactory bulbs from mice at P14–P21 were exposed and fluorescent reporters were directly visualized and imaged by a Nikon Ti2-E&CSU-W1 confocal microscope.

**In situ hybridization**. In situ hybridization was performed as previously described[2]. Full-length, anti-sense cRNA riboprobes labeled with digoxigenin were prepared to detect mRNA expression of *Taars*, *Olfr1507*, and *Olfr578* in single-color in situ hybridization experiments. The probe information and primers used to generate probes are summarized in Supplementary Tables 8 and 9, respectively.

For two-color in situ hybridization experiments, mixed riboprobes labeled with digoxigenin against all of the *Taar* genes (*Taar2*, *Taar3*, *Taar4*, *Taar5*, *Taar6*, *Taar7s*, *Taar8s*, and *Taar9*) or 8 *OR* genes (*Olfr644*, *Olfr578*, *Olfr1019*, *Olfr1034*, *Olfr145*, *Olfr1395*, *Olfr2*, *Olfr1507*) or 8 class I *OR* genes (*Olfr543*, *Olfr547*, *Olfr552*, *Olfr639*, *Olfr653*, *Olfr672*, *Olfr692*, *Olfr690*) and riboprobes labeled with fluorescein against GFP were used. Degenerate probes labeled with digoxigenin used to detect *OR* expression were generated as previously described[26]. The probe information and primers used to generate probes are summarized in Supplementary Tables 8 and 9, respectively. Images were taken using a Leica TCS SP8 confocal microscope. To quantify the number of cells expressing *Taars* or *ORs*, every 25th coronal section (14 μm thickness) throughout the MOE of mice at P14 was collected. At least five sections at similar anatomical positions in the MOE from mice with different genotypes were used to count positive cell numbers.

**Immunohistochemistry**. Coronal MOE sections (14 μm thickness) from P14 mice were fixed with 4% paraformaldehyde in phosphate-buffered saline (PBS) for 10 min at room temperature. The sections were washed with PBS three times (5 min each) and incubated with permeable buffer (0.3% Triton X-100 in PBS) containing 5% donkey serum for 30 min. Primary antibodies against TAAR4, TAAR5, TAAR6 (homemade), caspase-3 (Cell Signaling, 9661), GFP (Abcam, ab13970), and tdTomato (Takara, 632496) were used at 1:5000, 1:5000, 1:1000, 1:500, 1:1000, and 1:500 dilution, respectively, in incubation buffer (1% bovine serum albumin, 0.01% sodium azide, 0.3% Triton X-100 in PBS). Primary antibody incubations were performed at 4 °C for two overnights. The sections were then rinsed three times (5 min each) in PBS and incubated at 37 °C for 30 min with different fluorophore-conjugated secondary antibodies (diluted at 1:1000), including Donkey Anti-Chicken IgY conjugated to Alexa Fluo 488 (Jackson ImmunoResearch, 703-545-155), Donkey Anti-Rabbit IgG conjugated to Alexa Fluo 488 (Jackson Immu-noResearch, 711-545-152), and Donkey Anti-Guinea Pig IgG conjugated to Cy3 (Jackson ImmunoResearch, 706-165-148). Slides were rinsed three times (5 min each) and coverslipped using mounting medium containing DAPI (South-ernBiotech). Images were taken using a Leica TCS SP8 confocal microscope.

**Phylogenetic tree building**. The selected species for phylogenetic tree building are mammals that have the predicted *Taar* gene sequences and at least one of the two potential TAAR enhancers. The phylogenetic tree was built using TimeTree (http://www.timetree.org/)[52].

**Nucleotide percent identity plot**. Using NCBI Genome Data Viewer (https://www.ncbi.nlm.nih.gov/genome/gdv/), we compared mouse genome sequence

ranging from *Taar1* to *Taar7b* with that in 8 Glires, 21 EuarchontoGlires, 40 Placental mammals, and 60 Vertebrates by phyloP. The display style was chosen as line graph with smooth curve. The selected species and phylogenetic relationships are shown in Supplementary Fig. 3a. Next, the 991 bp TAAR enhancer 1 (mm10, chr10: 23,930,200-23,931,190) and 862 bp TAAR enhancer 2 (mm10, chr10: 23,987,501–23,988,362) sequences were separately blasted against the latest genome assembly of each species from NCBI, with the BLAST algorithm optimized for somewhat similar sequences (blastn) for more hits. The aligned sequences with the adjacent *Taar* gene sequences were downloaded for further analysis. The sequences for global alignment were extracted from the entire region of the *Taar* cluster. This approximately 295 kb mouse genomic region was compared with the same region in other species by using the web-based VISTA program (http://genome.lbl.gov/vista/mvista/submit.shtml)[53]. The local alignment of TAAR enhancer 1 and TAAR enhancer 2 sequences from different species were performed using the same program with the default settings of Shuffle-LAGAN alignment program and VISTA parameters. The setting of conservation identity was changed to 70% and that of calculation window and minimal conserved width was changed to 100 bp. The information of compared regions and selected genome databases is listed in Supplementary Table 1.

**Motif analysis**. Motif analysis of Lhx2, Ebf, and Lhx2/Ebf co-bound composite sites on the two TAAR enhancers was performed with HOMER (version 4.9). The motif position weight matrices were obtained from Lhx2 and Ebf ChIP-seq data on mature OSNs[16,17]. The Lhx2/Ebf co-bound composite sites with score >5 were retained.

**Reporter transgenic zebrafish**. Sequences of the two TAAR enhancers and the Sifnos OR enhancer were amplified from mouse genomic DNA using the following primers containing the restriction enzyme sites: 5'-CAGATGGGCCCTCGAGAA TGCACCAGTGCTCGTTGTG-3' (TAAR enhancer 1 forward), 5'-TAGAGTCGA GAGATCTTGTATGTAGCTGATGTCAGTATCTAGC-3' (TAAR enhancer 1 reverse), 5'-TAGAGTCGAGAGATCTTGTATGTAGCTGATGTCAGTATCTAG C-3' (TAAR enhancer 2 forward), 5'-TAGAGTCGAGAGATCTGACCAGCAGA TGAAGAAAG-3' (TAAR enhancer 2 reverse), 5'-CAGATGGGCCCTCGAGCAC CCCCAAGGGATTCAATG-3' (Sifnos enhancer forward), and 5'-TAGAGTCGAG AGATCTATAACTTGCTTCAAGACATGTG-3' (Sifnos enhancer reverse). PCR products were cloned into the E1B-GFP-tol2 vector via XhoI and BglII restriction sites[54]. The cloned plasmids were purified by Qiagen purification kit and were then injected into one-cell-stage zebrafish oocytes at 40 ng/μl together with 30 ng/μl Tol2 transposase mRNA. GFP expression was inspected at 24–48 high-power fields. Only embryos having at least one GFP-positive OSN were counted as positive embryos.

**Statistics**. Statistical analysis was performed using R (version 3.5.1) and GraphPad Prism (version 7.0a and 8.4.0). We calculated $p$ values by Wald test (two-sided) in DiffBind package for differential peak analysis in ATAC-seq data (Fig. 2b, d), by one-way analysis of variance and post hoc Tukey's test for multiple group comparison (Figs. 4f, h, 5f, 6f, and 8c and Supplementary Figs. 5d, e, 6e, f, and 7d), by the likelihood ratio test and Wald test (two-sided) in DESeq2 package for DEG analysis in RNA-seq data (Figs. 4c, d, 5c, d, and 6c, d and Supplementary Figs. 3a, 5a, b, 6a, b, and 7a, b), and by two-sided Fisher's exact test (Fig. 7c, e). In all the figures, $p$ values or $q$ values are denoted as *<0.05, **<0.01, ***<0.001.

**Accession codes**. The raw and processed sequencing data were deposited in the Gene Expression Omnibus with the accession number GSE163778.

**Reporting summary**. Further information on research design is available in the Nature Research Reporting Summary linked to this article.

## Data availability
The data supporting the findings of this study are included within the article and its Supplemental files. Source data are provided with this paper.

## Code availability.
The scripts for the analysis are available upon request. Source data are provided with this paper.

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

## Acknowledgements

We thank the Bob Datta laboratory at Harvard Medical School for kindly providing *Omp-ires-GFP* mouse line, the Marius Hoener laboratory at Roche Innovation Center Basel for sharing *Taar2-9* cluster knockout mouse line, the Stavros Lomvardas laboratory, and the Kevin Monahan laboratory for sharing RNA-seq data of Ldb1 and Lhx2 knockout mice, H3K9me3 ChIP-seq data, and HOMOER motif matrices. We would like to thank Wanxin Zeng from the Xiajing Tong laboratory at ShanghaiTech University for help on whole-mount imaging, Xiaocui Zhang for help on FACS experiments, Jason Buenrostro for advice on ATAC-seq, Matthew Lorincz and Julie Brind'Amour for advice on ULI-ChIP-seq, and Stavros Lomvardas for helpful discussion. We also want to thank members of the Li laboratory for critical reading of the manuscript. Computational analyses were performed on the Orchestra and O₂ High-performance Computer Cluster at Harvard Medical School. Images were taken in Core Facility of Basic Medical Sciences, Shanghai Jiao Tong University School of Medicine and the Molecular Imaging Core Facility (MICF) at School of Life Science and Technology, ShanghaiTech University. This work was supported by National Natural Science Foundation of China (31771154 and 31970933 to Q.L.), Shanghai Brain-Intelligence Project from the Science and Technology Commission of Shanghai Municipality (18JC1420302), Shanghai Pujiang Program (17PJ1405400 to Q.L.), Program for Young Scholars of Special Appointment at Shanghai Institutions of Higher Learning (QD2018017 to Q.L.), Innovative research team of high-level local universities in Shanghai, Fundamental Research Funds for the Central Universities (Shanghai Jiao Tong University, 17X100040037 to Q.L.), and an NIH Director's

Pioneer Award (DP1 CA186693 to X.S.X.). L.T. was supported by an HHMI International Student Research Fellowship. Work at the G.B.'s laboratory was supported by grant numbers 1R01DC013561 and P30GM103410.

## Author contributions

A.F., W.W., L.T., and Q.L. designed experiments, analyzed data, and wrote the manuscript; L.T., X.S.X., G.B., S.D.L., and Q.L. edited and revised the manuscript. A.F., W.W., L.T., H.B., Y.K., and C.Y. performed morphological, histological, RNA-seq, ATAC-seq, and animal experiments; C.T., X.H., and H.Y. generated enhancer knockout and transgenic mouse lines; Z.X. performed phylogenetic analysis of TAAR enhancers; C.R., K.J.H., and F.E. performed zebrafish embryo injection and imaging experiments; M.J., G.H., M.T., and G.B. made the antibodies against TAAR4, TAAR5, and TAAR6 prior to publication[26] and generated the *Taar5-ires-Cre* and *Taar6-ires-Cre* mice.

## Competing interests

The authors declare no competing interests.
