## [Peer Review File · Nature Communications]

Reviewers' Comments:

Reviewer #1:

Remarks to the Author:

The manuscript by Li and colleagues investigates the cis regulatory elements responsible for the expression of the clustered Trace amine-associated receptors (TAARs) in olfactory sensory neurons (OSNs). Using genetically encoded fluorescent reporters the authors successfully isolate TAAR-expressing OSNs which they analyze by RNA-seq, ChIP-seq and ATAC-seq. RNA-seq confirms these OSNs express TAARs and reveals the TAAR choice is less stable than olfactory receptor (OR) choice, since there is abundant detection of most TAARs in this population. ChIP-seq revealed that the TAAR cluster becomes enriched for H3K9me3 in TAAR-expressing OSNs, unlike OR-expressing OSNs where the TAAR cluster is heterochromatin-free. Finally, ATAC-seq uncovered the location of two putative enhancer elements which the authors analyzed by loss and gain of function experiments. As their genetic experiments demonstrate, deletion of individual enhancers has modest effects that influence the expression of overlapping TAAR repertoires, while the double enhancer deletion abolishes the expression of every OSN-expressed TAAR gene. Finally, with clever complementation experiments the authors demonstrate that TAAR enhancers function as cis enhancers. In parallel studies where the authors generated transgenic reporter mice, it becomes apparent that the two TAAR enhancers contain sufficient information to restrict enhancer activity in TAAR-expressing OSNs, suggesting that they are both bound transcription factor combinations that are expressed specifically in the TAAR-expressing OSN lineage.

The manuscript is well written and provides significant advancement to our understanding of TAAR gene regulation and the experiments are rigorous and the results are convincing. Thus, I am in favor of accepting this manuscript for publication.

Editorial suggestions: Although the main finding is the identification and characterization of the two enhancers, I find also important the demonstration that TAARs become heterochromatic in TAAR-expressing OSNs. In their discussion, the authors may want to entertain the possibility that heterochromatin formation on the TAAR cluster contributes not only to the relocation from the nuclear lamina but also to the recruitment to the OR compartment and the eventual interaction with other OR enhancers, as recently demonstrated for ORs (Bashkirova et al 2020 bioRxiv). Moreover, I would like to emphasize that the authors should be a bit more conservative when they proclaim the two enhancers as cis enhancers based only on genetic studies. As previous experiments showed the roles of OR enhancers are very complex and the term "cis only enhancers" is unfortunate, to say the least; Monahan and colleagues (Nature, 2019) showed that OR enhancer deletions prevent OR clusters from being recruited to OR compartments, while it also prevents recruitment of other OR enhancers in trans. Thus, although by the strict definition of the term OR enhancers act only in cis, analysis of the effects of enhancer deletion in nuclear architecture revealed that these enhancers act as ZIP codes that enable trans genomic interactions. Something similar may be happening with the TAAR enhancers, where in TAAR-expressing cells they may be recruited to OR enhancer hubs, consistent with the fact that OR enhancers retain ATAC signal in these TAAR-expressing OSNs. This would be also consistent with the effects of Lhx2 and Ldb1 deletion in TAAR expression. I am not asking for a HiC analysis in the FAC-sorted TAAR-expressing cells, but I am just suggesting the possibility that the chosen TAAR is also recruited to Greek Island hubs in a fashion that is dependent on the cis action of the two TAAR enhancers. In other words, I am suggesting the exciting possibility that the TAAR locus is hijacking the endogenous, common, apparatus used for singular OR gene choice.

Reviewer #2:

Remarks to the Author:

How olfactory sensory neurons (OSNs) achieve monogenic and monoallelic expression of odorant receptors is an important and interesting question. While the vast majority of OSNs express the canonical G-protein coupled odorant receptors (ORs), a subset of OSNs express trace amine-associated receptors (TARRs). Compared to the canonical ORs, much less is known about regulation of

receptor gene choice and expression in TAAR expressing OSNs. Here Qian Li and his colleagues addresses this question using an impressive array of cutting-edge molecular and genetic approaches. They identified two evolutionarily conserved TAAR enhancers and demonstrated their necessity in Taar expression: deletion of either one decreases the expression probabilities of multiple Taar genes, and deletion of both completely eliminates TAAR OSNs. The authors then showed sufficiency of the two enhancers in driving Taar gene expression and demonstrated that enhancer 1 operates in cis. This work significantly advances our understanding of Taar gene choice and expression regulation in OSNs. The experiments are well designed, the data are in high quality, the statistical analysis is appropriate, and the conclusions are sound. This is an elegant piece of work and a well-written manuscript.

I have only minor comments and suggestions:

1. Fig. S1A, the font size for the axis marks is too small to read.
2. Fig. S2, "ONSs" should be "OSNs" (a total of four on the axis label).
3. Fig. S3, for Taar8c, is the difference between the wild-type and Ldb1 knockout not significant? It would be clearer if n.s. is added to the figure.
4. For Fig. 4C, D, and Fig. 5C, D, it would be helpful to label the blue dots with Taar gene number (e.g. Taar 5, 7b, etc).
5. Page 21, Line 1, "observed a few glomeruli that were positive for GFP or tdTomato alone (Figure 7G)". The enlarged pictures showed only co-expressed glomeruli. Use arrows pointing to single labeled glomeruli in the left panel will make this point clearer.
6. Page 22 line 4-10. The description of the four genotypes is redundant since they are clearly shown in Fig. 8A. This part can be simplified. For example, "(2) genotype 2: one allele lacking TAAR enhancer 1 and the other allele intact", etc.
7. Figure 7 title "TAAR enhancer 1 possesses ...". In fact, data for both enhancers are included. Suggest revising the title to "TAAR enhancer 1 and 2 possess".
8. Not all authors are mentioned in author contributions.

By Minghong Ma

Reviewer #3:

Remarks to the Author:

Review comments: NCOMMS-20-31472

In the mouse olfactory epithelium (MOE), the majority of olfactory sensory neurons (OSNs) expresses odorant receptors (ORs) to detect odorous chemicals in the environments. Besides them, trace amine-associated receptors (TAARs) consisting a distinct olfactory receptor subfamily are expressed in a subpopulation of OSNs, some of which have been demonstrated to detect predator signal and social cues to induce innate olfactory behaviors. The molecular mechanisms by which individual OSNs select to express a single OR gene from up to 1,100 OR genes have been gradually elucidated over the past two decades through the identification of cis-regulatory elements and the description of epigenetic modifications including the formation of OR compartments and enhancer hubs. However, those of TAAR genes remain largely unclear.

The present work by Fei and coworkers, for the first time, identifies two cis-elements that coordinately regulate TAAR gene expression by using a combinations of epigenetic analyses, sequence comparison approaches and mouse genetic studies including loss-of-function analyses and transgenic reporter assays, and adds new insights into the emerging picture of receptor gene regulation in OSNs.

Although the cooperative action of two or more cis-elements (enhancers) have been expected and demonstrated through the previous studies on the H and P elements by Sakano and the Mombaerts groups and the recent interchromosomal cis/trans-interactions of class II enhancers (Greek islands), by the Stavros group, the present study clearly demonstrates this experimentally and provides clear evidence for the strict cis-regulation by these enhancers that does not extend to OR genes.

The experiments and analyses were generally conducted with adequate care and accuracy. There are

several issues that need to be addressed to improve the quality of the work.

Major points:

1) page 7 "Enrichment of TAAR OSNs". The results of TAAR gene expression patterns in TAAR5-ires-Cre/GFP-positive OSNs and in TAAR6-ires-Cre/ZsGreen-positive OSNs (Fig 1C) are very interesting. Contrary to the expectations, various TAAR genes were expressed in addition to the knocked in TAAR genes.

In TAAR5-ires-Cre mice, TAAR5 expression level was very low, instead, TAAR4 expression was dominant. Are there any relations to the frequency (probability) of TAAR gene choice in normal mice? Please present the each TAAR gene expression data (TPM) in OSNs (OMP-GFP mice) or cite relevant references. In addition, when the "ventral" TAAR6 expressing cells were tagged, both ventral and dorsal TAAR genes were expressed. Are there any spatial effects?

These interesting phenomena were rarely described and discussed. The authors attributed these phenomena to "highly frequent receptor switching". I do think this is one possibility, however, multiple OR genes are coexpressed in a single OR-OSN in the early stage of differentiating OR-OSNs (Hanchate et al., Science 2015). It would be fair to also mention this possibility.

Another possibility is that, contrary to the OR loci, the TAAR cluster may be devoid of heterochromatin modifications in the precursor and/or immature stages allowing multiple TAAR genes expression, and the level of H3K9me3 may increase as TAAR-OSNs mature. H3K9me3 ChIP-seq data of precursor/immature OSNs is required (related to the comment #2).

2) Figure 1E. The differential heterochromatin modifications of the TAAR cluster in TAAR-OSNs and in OR-OSNs are new findings and interesting. However, the resolution of the ChIP data was too low. Please provide high resolution data so that readers can see the level of H3K9me3 of each TAAR gene locus. Related to the above comment #1, the data must be helpful to explain the results of Fig. 1C by examining whether the TAAR5 locus is marked with H3K9me3 in TAAR5-ires-Cre/GFP-positive-OSNs, and the TAAR4 is devoid of H3K9me3 modification. Please present both data of TAAR5+ and TAAR6+, which is also helpful to discuss the hypothesis of the involvement of LSD1 in epigenetic escape from heterochromatin modifications (page 9, line 9-).

Then, one key question has been raised, is the level of H3K9me3 in the TAAR locus increased during the differentiation of TAAR-OSNs (the TAAR locus is devoid of the heterochromatin modifications in the early differentiating stage)? or is the level of H3K9me3 in the TAAR locus decreased in OR-OSNs? Please make this point clear.

3) page 11, line3-8, "The co-existence of putative TAAR enhancers and OR enhancers in TAAR OSNs indicates that they may form an enhancer hub to facilitate Taar gene choice....". I think these sentences are logically strange. The ATAC-seq only represents chromatin accessibility but not indicates any chromatin interactions. It seems to me these sentences are over-speculations. To make this point clear, Hi-C analyses are required. Considering the objective of this study to identify TAAR enhancers, the ATAC-seq data of OR enhancers (Figure 2E, 2F) are no necessary, rather it dilute the purpose of this study when presented in Figure2. I would suggest deleting them.

4) Related to the above comments #3, analyses of the conserved motifs in TAAR enhancers are mandatory. Are there any conserved sequence motifs in the TAAR enhancer sequences (e.g., Lhx2-binding motif, Ebf-binding(O/E-like) motif, and the Greek island composite motif)? In addition, both the local cis-acting OR enhancers and the formation of interchromosomal enhancer hubs are required for the stable and high level expression of a single OR gene. Deletion of the cis-element (H, P, Lipsi, for example) resulted in abolishment of the liked OR genes expression and the formation of enhancer hub on a expressed OR gene locus, which are facilitated by the dimerization ability of Ldb1. At the same time, Ldb1 is required for the transcriptional activity of Lhx2, as Lhx2 do not function as a transcription factor without LIM-binding proteins. Thus, it is also possible that the reduced expression of TAAR genes in Ldb1 KO is caused by the loss of transcriptional activity Lhx2 function (local cis-acting activity) as well as failure of making up enhancer hubs.

5) page 14 - 17, the description of the results of TAAR enhancer 1 knockout and enhancer 2 knockout is confusing.

For example, the authors say "the numbers of TAAR OSNs expressing Taar3, Taar4, Taar5...were dramatically decreased in TAAR enhancer 2 knockout mice" in page 16, line 17-19, but in the next page, "cells expressing Taar3 and Taar5 were completely abolished in TAAR enhancer 2 knockout mice" (page 17, line 1-2).

In page 14, line 18 "it tended to decrease for Taar7f and to increase for Taar7e (in TAAR enhancer 1 KO)." but in page 17, line 11-12 " (Taar7f) were not changed in TAAR enhancer 1 knockout mice."

Please rewrite these sections to ensure that there are no inconsistencies between the descriptions of the results. Also, please be consistent in the wording (there are so many terms, significantly, largely, dramatically, completely, tended to, not changed...).

In addition, please present a table summarizing the results of the RNA-seq and ISH analyses (expression level, fold of change of expression level, the number of cells, fold of change of the number, and their statistical data of each Taar gene in TAAR enhancer 1 KO, TAAR enhancer 2 KO, and double KO) to compare the effects of the deletion of enhancer(s) on Taar gene expression.

6) In Figure 6, the RNA-seq data and quantification data of apoptotic cells are missing. Please provide them. The RNA-seq data is necessary to demonstrate "no effects" on the expression OR genes and other genes outside the cluster. The number of apoptotic cells should be presented to examine and discuss the arrest of neuronal differentiation of TAAR-OSNs.

7) page 19, ISH analyses of the coexpression of reporter genes and OR genes. The authors used OR mixed probes. Please describe the character of OR genes used in the mixed probe, e.g., class I or class II, dorsal or ventral OR. In this assay, the authors argued specific expression of reporter genes in TAAR-OSNs judged by coexpression rate. However, the mixed probes are composed of 8 OR genes out of 1,100 OR genes (probably the authors chose dorsal OR genes, if so, out of ~250 genes). Considering the total number of OR genes, I do not think the authors can simply conclude the specific expression of reporter genes in TAAR-OSNs. RNA-seq analyses of FACS sorted OSNs provide convincing conclusions.

In addition, I do not understand the reason why coexpression rates of OR degenerate primers were much lower than those of OR mixed probe. Please explain. Does the degenerate probe detect class I OR genes? As the OR mixed probe includes some class I OR probes, is there any possibility that TAAR-enhancer reporters coexpress class I OR with high frequency?

8) page 21, line 5-10 and other, similar expression levels (number of cells?) of Taar genes in heterozygous TAAR enhancer 1, 2 and 1&2 knockout mice compared to their wild type littermates. However, authors only provided the normalized data. One cannot evaluate the total number of cells from the ratio between knockout and control of each Taar gene. For example, in the manuscript the number of Taar3 expressing OSNs significantly decreased in TAAR enhancer 1 and enhancer 3 knockout mice. If Taar3 expressing OSNs are dominant in the TAAR-OSN population, that largely affects the total number of TAAR OSNs. Please provide all quantification data (the number of cells) as supplementary table together with statistical analyses and compare the total number of TAAR-positive cells.

Minor points:

1) Regarding to figure representations, please take into account the color barrier. Please replace red color with magenta or color blind friendly one.

2) page 7, line 19-20, "approximately 70-80% of which are composed of OR-expressing mature OSNs". How did the authors estimate the population? Please display the data, or put appropriate references.

3) page 12, line 10-11, "loss of TAAR enhancer 2 in dog, sea lion, seal,,,,". The VISTA plot data is missing. Please display all data described in the manuscript.

4) page 14, line 11-14, " No obvious differences in the numbers of cells expressing Taar7s were detected....". I think that ISH data using Taar7 coding probes is neither conclusive nor necessary (especially the expression level of Taar7e increased in the TAAR enhancer 1 KO). Please delete them from the manuscript and revise Figure 4E, 4F, 5E, 5F. The data acquired using specific probes are sufficient.

5) page 18, line 7, "distinct groups of Taar genes" would mislead the readers. Some Taar genes expression were decreased in both enhancer KO mice. the effects of TAAR enhancers overlap each other. Please reword.

6) The section of "TAAR enhancers are sufficient to drive reporter expression in the TAAR OSNs". In page 19, line 12, it is my personal opinion that it is not appropriate to include "data not shown". This is an interesting and experimentally important data, which in my opinion is well worth presenting. Because some Taar genes are expressed ventrally. Also please consider displaying both the MOE (lateral view) and the OB (both lateral and top) . In addition, Figure 7G (the left panel) is difficult to see. Please present both single color and merge images.

7) page 20, line 14, "to examine the coordination between two TAAR enhancers". There is no doubt that endogenous TAAR enhancers coordinate to regulate Taar genes in cis. However, in the transgenic line, TAAR enhancers-GFP/tdTomato transgenes are integrated randomly into chromosomes (different position, different copy number). Apparently, two TAAR enhancers "cooperate" to regulate Taar7d and 7e, because the deletion of each enhancer did not change these expression but the deletion of both abolished them. How the authors examine the coordination between two differentially integrated transgenes? Do GFP single or tdTomato single positive OSNs reflect that only one of two endogenous enhancers is active in them? I do not think the conclusion can be drawn from the results of this experiment. Coexpression rate of each Taar genes with reports would be more helpful.

8) page 22, line 22 "thus stabilize the cell fate decision of the TAAR subsystem". I do not understand this sentence. From which experiment can this conclusion be drawn? The authors mentioned that deletion of enhancers arrest the differentiation of TAAR-OSNs. In other words, the cell fate is not destabilized by the deletion of enhancers. Please reword.

9) Figure 8C. Are there significant differences between genotype 1 versus 2, genotype 1 versus 3? Please also mention and discuss the residual Taar 6 expression in genotype 4. Are there any trans-interaction between TAAR enhancer 1 and Taar6 genes? Only Taar6 was unchanged between genotype 1 versus 2, whereas Taar 2, 3 and 5 were tended to decrease (or were decreased).

10) Supplementary Figures 5, 6. Only provide all DEG data (TAAR enhancer 1, 2 and 1&2 KO) as Supplementary Table.

11) This is out of the objective of the present study, but please state the effect of TAAR enhancers knockout on Taar 1 expression in brain, if possible.

12) page43, Figure 7 legend title. Because the authors demonstrated expression patterns of both TAAR enhancer 1 and 2, the title should be "TAAR enhancers" of "TAAR enhancer 1 and enhancer 2".

13) In Figure 7 legend (page 43, line22) "The number of GFP-positive and tdTomato-positive founders among the total analyzed lines". In the Results, the authors mentioned to keep one transgenic line for each enhancer and analyze them. I think "among the total analyzed founders/lines" is correct. Please confirm.

14) The Method is not sufficient. Please provide the following information (I recommend provide detailed information as Supplementary Tables
ISH probe information (name, genbank ID, nucleotide position, or references)
plasmid information (name, Addgene#, reference) for CRISPR-Cas9 gRNA & SpCas9 templates and for PiggyBac transposon plasmid
primer sequence for Enhancer#1 or #2 knockout genotyping
Deposit ID of the metadata analyzed in this study.

REVIEWER COMMENTS

Reviewer #1 (Remarks to the Author):

The manuscript by Li and colleagues investigates the cis regulatory elements responsible for the expression of the clustered Trace amine-associated receptors (TAARs) in olfactory sensory neurons (OSNs). Using genetically encoded fluorescent reporters the authors successfully isolate TAAR-expressing OSNs which they analyze by RNA-seq, ChIP-seq and ATAC-seq. RNA-seq confirms these OSNs express TAARs and reveals the TAAR choice is less stable than olfactory receptor (OR) choice, since there is abundant detection of most TAARs in this population. ChIP-seq revealed that the TAAR cluster becomes enriched for H3K9me3 in TAAR-expressing OSNs, unlike OR-expressing OSNs where the TAAR cluster is heterochromatin-free. Finally, ATAC-seq uncovered the location of two putative enhancer elements which the authors analyzed by loss and gain of function experiments. As their genetic experiments demonstrate, deletion of individual enhancers has modest effects that influence the expression of overlapping TAAR repertoires, while the double enhancer deletion abolishes the expression of every OSN-expressed TAAR gene. Finally, with clever complementation experiments the authors demonstrate that TAAR enhancers function as cis enhancers. In parallel studies where the authors generated transgenic reporter mice, it becomes apparent that the two TAAR enhancers contain sufficient information to restrict enhancer activity in TAAR-expressing OSNs, suggesting that they are both bound transcription factor combinations that are expressed specifically in the TAAR-expressing OSN lineage. The manuscript is well written and provides significant advancement to our understanding of TAAR gene regulation and the experiments are rigorous and the results are convincing. Thus, I am in favor of accepting this manuscript for publication.

Editorial suggestions: Although the main finding is the identification and characterization of the two enhancers, I find also important the demonstration that TAARs become heterochromatic in TAAR-expressing OSNs. In their discussion, the authors may want to entertain the possibility that heterochromatin formation on the TAAR cluster contributes not only to the relocation from the nuclear lamina but also to the recruitment to the OR compartment and the eventual interaction with other OR enhancers, as recently demonstrated for ORs (Bashkurova et al 2020 bioRxiv). Moreover, I would like to emphasize that the authors should be a bit more conservative when they proclaim the two enhancers as cis enhancers based only on genetic studies. As previous experiments showed the roles of OR enhancers are very complex and the term “cis only enhancers” is unfortunate, to say the least; Monahan and colleagues (Nature, 2019) showed that OR enhancer deletions prevent OR clusters from being recruited to OR compartments, while it also prevents recruitment of other OR enhancers in trans. Thus, although by the strict definition of the term OR enhancers act only in cis, analysis of the effects of enhancer deletion in nuclear architecture revealed that these enhancers act as ZIP codes that enable trans genomic interactions. Something similar may be happening with the TAAR enhancers, where in TAAR-expressing cells they may be recruited to OR enhancer hubs, consistent with the fact that OR enhancers retain ATAC signal in these TAAR-

expressing OSNs. This would be also consistent with the effects of *Lhx2* and *Ldb1* deletion in TAAR expression. I am not asking for a HiC analysis in the FAC-sorted TAAR-expressing cells, but I am just suggesting the possibility that the chosen TAAR is also recruited to Greek Island hubs in a fashion that is dependent on the *cis* action of the two TAAR enhancers. In other words, I am suggesting the exciting possibility that the TAAR locus is hijacking the endogenous, common, apparatus used for singular OR gene choice.

We thank the reviewer for the positive comments. We agree with the reviewer's points about heterochromatin decoration on the *Taar* cluster, *cis*-acting TAAR enhancer, and formation of the enhancer hub. In the revised manuscript, we carefully revised the Discussion in page 28 line 13-15, page 30 line 4-8, and page 30 line 10-15 according to the reviewer's comments. The revised text is shown below:

"However, the two TAAR enhancers could also mediate *trans* genomic interactions to form a TAAR enhancer hub to facilitate *Taar* gene transcription, in analogous to OR enhancers (Monahan et al., 2019, Nature; Markenscoff-Papadimitriou et al., 2014, Cell) (see below for further discussion)."

"It is very likely that the heterochromatin formation on the *Taar* cluster contributes to the relocation of *Taar* genes from the nuclear lamina in TAAR OSNs, which is consistent with the recent finding that the levels of heterochromatin are strongly correlated with the frequency of Hi-C contacts among the *OR* clusters (Bashkirova et al., 2020, bioRxiv)."

"The two TAAR enhancers could act in *trans* to form a TAAR enhancer hub with OR enhancers based on the finding that OR enhancers are also open in TAAR OSNs. This together with the distinct heterochromatin decoration on the *Taar* cluster raise an interesting possibility that the TAAR olfactory subsystem is hijacking the pre-existed apparatus for singular *OR* gene choice by simply developing the two TAAR enhancers."

Reviewer #2 (Remarks to the Author):

How olfactory sensory neurons (OSNs) achieve monogenic and monoallelic expression of odorant receptors is an important and interesting question. While the vast majority of OSNs express the canonical G-protein coupled odorant receptors (ORs), a subset of OSNs express trace amine-associated receptors (TARRs). Compared to the canonical ORs, much less is known about regulation of receptor gene choice and expression in TAAR expressing OSNs. Here Qian Li and his colleagues addresses this question using an impressive array of cutting-edge molecular and genetic approaches. They identified two evolutionarily conserved TAAR enhancers and demonstrated their necessity in *Taar* expression: deletion of either one decreases the expression probabilities of multiple *Taar* genes, and deletion of both completely eliminates TAAR OSNs. The authors then showed sufficiency of the two enhancers in driving *Taar* gene expression and demonstrated that enhancer 1 operates in *cis*. This work significantly advances our understanding of *Taar* gene choice and expression regulation in OSNs. The experiments are well

designed, the data are in high quality, the statistical analysis is appropriate, and the conclusions are sound. This is an elegant piece of work and a well-written manuscript.

We thank the reviewer for the positive remarks. Please see our responses to the reviewer's minor points.

I have only minor comments and suggestions:

1. Fig. S1A, the font size for the axis marks is too small to read.

We increased the font size in Supplementary Figure 1A.

2. Fig. S2, "ONs" should be "OSNs" (a total of four on the axis label).

We corrected the errors in Supplementary Figure 2.

3. Fig. S3, for *Taar8c*, is the difference between the wild-type and *Ldb1* knockout not significant? It would be clearer if n.s. is added to the figure.

The expression level of *Taar8c* in the *Ldb1* knockout is not significantly different from that in the wild type. We added the n.s. label and the relevant figure legend in Supplementary Figure 3.

4. For Fig. 4C, D, and Fig. 5C, D, it would be helpful to label the blue dots with *Taar* gene number (e.g. *Taar* 5, 7b, etc).

We labeled the blue dots with the *Taar* gene names in Figure 4C, 4D, 5C, 5D, and new 6C, 6D.

5. Page 21, Line 1, "observed a few glomeruli that were positive for GFP or tdTomato alone (Figure 7G)". The enlarged pictures showed only co-expressed glomeruli. Use arrows pointing to single labeled glomeruli in the left panel will make this point clearer.

We added arrows and arrow heads to mark the single labeled glomeruli in Figure 7G and revised the figure legend accordingly.

6. Page 22 line 4-10. The description of the four genotypes is redundant since they are clearly shown in Fig. 8A. This part can be simplified. For example, "(2) genotype 2: one allele lacking *TAAR* enhancer 1 and the other allele intact", etc.

We changed the text in page 25 line 15-20 following the reviewer's suggestion.

7. Figure 7 title "*TAAR* enhancer 1 possesses ...". In fact, data for both enhancers are included. Suggest revising the title to "*TAAR* enhancer 1 and 2 possess".

We changed the Figure 7 title to "*TAAR* enhancer 1 and enhancer 2 possess *TAAR* OSN-specific enhancer activity".

8. Not all authors are mentioned in author contributions.

We revised the Author contributions section and included all of the authors. Thanks for the note.

By Minghong Ma

Reviewer #3 (Remarks to the Author):

Review comments: NCOMMS-20-31472

In the mouse olfactory epithelium (MOE), the majority of olfactory sensory neurons (OSNs) expresses odorant receptors (ORs) to detect odorous chemicals in the environments. Besides them, trace amine-associated receptors (TAARs) consisting a distinct olfactory receptor subfamily are expressed in a subpopulation of OSNs, some of which have been demonstrated to detect predator signal and social cues to induce innate olfactory behaviors. The molecular mechanisms by which individual OSNs select to express a single OR gene from up to 1,100 OR genes have been gradually elucidated over the past two decades through the identification of cis-regulatory elements and the description of epigenetic modifications including the formation of OR compartments and enhancer hubs. However, those of TAAR genes remain largely unclear.

The present work by Fei and coworkers, for the first time, identifies two cis-elements that coordinately regulate TAAR gene expression by using a combinations of epigenetic analyses, sequence comparison approaches and mouse genetic studies including loss-of-function analyses and transgenic reporter assays, and adds new insights into the emerging picture of receptor gene regulation in OSNs. Although the cooperative action of two or more cis-elements (enhancers) have been expected and demonstrated through the previous studies on the H and P elements by Sakano and the Mombaerts groups and the recent interchromosomal cis/trans-interactions of class II enhancers (Greek islands), by the Stavros group, the present study clearly demonstrates this experimentally and provides clear evidence for the strict cis-regulation by these enhancers that does not extend to OR genes.

The experiments and analyses were generally conducted with adequate care and accuracy. There are several issues that need to be addressed to improve the quality of the work.

We want to thank the reviewer for the positive comments on our findings. We also appreciate his/her critical thinking when evaluating our paper. We have performed new experiments and rewritten some sections to address the reviewer's concerns. In a short summary, the main changes in the revised manuscript include:

- 1) careful discussion on the interesting phenomena of multiple *Taar* gene expression in the *Taar5/6-ires-Cre* mice and H3K9me3 decoration on the *Taar* cluster in difference cell types across the OSN lineage;
- 2) analysis of the conserved motifs, including Ldb1-binding motif, Lhx2-binding motif, Ebf-binding motif, and Lhx2/Ebf composite motif in the two TAAR enhancers;

- 3) performing RNA-seq and caspase-3 staining on the TAAR enhancer 1 & 2 double knockout mice to elucidate the specific effects of TAAR enhancers on *Taar* gene regulation;
- 4) more careful characterization of the enhancer transgenic animals using RNA-seq of sorted reporter-positive cells and *in situ* analysis with individual *Taar* probes for coexpression rates with reporters;
- 5) providing Supplementary Table 2, 3, 4, 5, 6, 7, and 8 with detailed quantification data of *in situ* and RNA-seq data, and Supplementary Table 9, 10, and 11 with detailed information for Methods as the reviewer suggested. In addition, the raw data were provided as source data file.

We think these changes together significantly improve the quality of our work, and hope that the reviewer will agree.

Major points:

1) page 7 "Enrichment of TAAR OSNs". The results of TAAR gene expression patterns in TAAR5-ires-Cre/GFP-positive OSNs and in TAAR6-ires-Cre/ZsGreen-positive OSNs (Fig 1C) are very interesting. Contrary to the expectations, various TAAR genes were expressed in addition to the knocked in TAAR genes. In TAAR5-ires-Cre mice, TAAR5 expression level was very low, instead, TAAR4 expression was dominant. Are there any relations to the frequency (probability) of TAAR gene choice in normal mice? Please present the each TAAR gene expression data (TPM) in OSNs (OMP-GFP mice) or cite relevant references. In addition, when the "ventral" TAAR6 expressing cells were tagged, both ventral and dorsal TAAR genes were expressed. Are there any spatial effects?

These interesting phenomena were rarely described and discussed. The authors attributed these phenomena to "highly frequent receptor switching". I do think this is one possibility, however, multiple OR genes are coexpressed in a single OR-OSN in the early stage of differentiating OR-OSNs (Hanchate et al., Science 2015). It would be fair to also mention this possibility.

Another possibility is that, contrary to the OR loci, the TAAR cluster may be devoid of heterochromatin modifications in the precursor and/or immature stages allowing multiple TAAR genes expression, and the level of H3K9me3 may increase as TAAR-OSNs mature. H3K9me3 ChIP-seq data of precursor/immature OSNs is required (related to the comment #2).

We added a paragraph regarding the reviewer's question about expression of multiple *Taar* genes as below:

"Alternatively, transient coexpression of multiple *Taar* genes in immature TAAR OSNs revealed by two-color *in situ* hybridization in our previous study could also explain the phenomena (Tan et al., 2015, MSB). Moreover, we found that *Taar4* was dominantly expressed in *Taar5*⁺ OSNs, and that both ventral and dorsal *Taar* genes were expressed in *Taar6*⁺ OSNs (Figure 1C). This is consistent with a previous study analyzing coexpression of *lacZ* gene with *Taar* genes in *Taar5*^{lacZ/+} or *Taar6*^{lacZ/+} mice, where the *Taar5* or *Taar6* coding region was replaced by a *lacZ* gene in one allele (Yoon et al., 2015, PNAS)." in page 8 line 13-20 of the revised manuscript.

To rule out the presence of H3K9me3 in the TAAR cluster in precursor/immature OSNs, Dr. Stavros Lomvardas has kindly helped us to reanalyze his H3K9me3 ChIP-seq data from his recent preprint (Bashkirova et al., bioRxiv 2020), and found no H3K9me3 in any developmental stages that they studied: GBCs, INPs, iOSNs, and mOSNs. Therefore, we added a paragraph regarding the reviewer's question about the H3K9me3 mark in OSN precursors as below: "Furthermore, the *Taar* cluster are not marked by H3K9me3 in globose basal stem cells (GBCs), immediate neuronal precursors (INPs), immature OSNs (iOSNs), or mature OSNs (mOSNs) prepared from the whole MOE according to the recent ChIP-seq data (personal communication with Stavros Lomvardas and Lisa Bashkirova; data from Bashkirova et al., 2020, bioRxiv), indicating that the H3K9me3 decoration on the *Taar* cluster is absent throughout the OR OSN lineage. However, since we do not have genetic mouse lines to enrich GBC, INP, and iOSN subpopulations of the TAAR OSN lineage, we can only postulate that the H3K9me3 decoration on the *Taar* cluster arises in TAAR OSN progenitors." in page 9 line 19-22 and page 10 line 1-5 of the revised manuscript.

2) Figure 1E. The differential heterochromatin modifications of the TAAR cluster in TAAR-OSNs and in OR-OSNs are new findings and interesting. However, the resolution of the ChIP data was too low. Please provide high resolution data so that readers can see the level of H3K9me3 of each TAAR gene locus. Related to the above comment #1, the data must be helpful to explain the results of Fig. 1C by examining whether the TAAR5 locus is marked with H3K9me3 in TAAR5-ires-Cre/GFP-positive-OSNs, and the TAAR4 is devoid of H3K9me3 modification. Please present both data of TAAR5+ and TAAR6+, which is also helpful to discuss the hypothesis of the involvement of LSD1 in epigenetic escape from heterochromatin modifications (page 9, line 9–).

We respectfully argue that such high-resolution low-input ChIP-seq data is technically prohibited by the low cell number of our mice, and therefore is beyond the scope of this paper. In particular, obtaining ChIP-seq data with single-*Taar*-gene resolution is extremely challenging in our experiments for 3 reasons: (a) Only 1 of the 2 parental alleles will be active (and thus H3K9me3). Therefore, we would need to discern a quantitative difference of only 2 fold (50% H3K9me3 vs. 100%), which would be easily overwhelmed by technical noise. (b) The enriched reporter-positive cells are a mixture of TAAR OSNs expressing different TAAR family members, which would further reduce the signals. (c) Because each *Taar* gene is short (~1 kb), each allele of each single cell would contribute to at most ~8 molecules of nucleosomes. Given the low efficiency of ChIP-seq and our limited number of GFP-positive cells, we would need an unrealistic number of mice and labor to obtain meaningful data.

However, we agree with the reviewer about the importance of this point, and therefore have added a sentence "Note that limited by low cell numbers, our data does not have single-*Taar*-gene resolution." in page 9 line 18-19 of the revised manuscript.

Then, one key question has been raised, is the level of H3K9me3 in the TAAR locus increased during the differentiation of TAAR-OSNs (the TAAR locus is devoid of the

heterochromatin modifications in the early differentiating stage)? or is the level of H3K9me3 in the TAAR locus decreased in OR-OSNs? Please make this point clear.

Please see our response to Major Point 1.

3) page 11, line3-8, "The co-existence of putative TAAR enhancers and OR enhancers in TAAR OSNs indicates that they may form an enhancer hub to facilitate Taar gene choice....". I think these sentences are logically strange. The ATAC-seq only represents chromatin accessibility but not indicates any chromatin interactions. It seems to me these sentences are over-speculations. To make this point clear, Hi-C analyses are required. Considering the objective of this study to identify TAAR enhancers, the ATAC-seq data of OR enhancers (Figure 2E, 2F) are no necessary, rather it dilute the purpose of this study when presented in Figure2. I would suggest deleting them.

We thank the reviewer for the critical thinking. We agree that the direct evidence for enhancer hub formation is lacking in this study. And Hi-C experiments are required to draw the conclusion. Actually reviewer #1 is very excited about the possibility that the TAAR enhancers are recruited to Greek Island hubs to regulate *Taar* gene expression (see comments from reviewer #1). Motif analysis revealed Ldb- and Lhx2-binding motifs in the two TAAR enhancers (see our responses to Major Point 4). After thinking carefully of comments from both reviewers, we kept the results in Figure 2, and revised the text in page 11 line 18-22 and page 12 line 1-13 to avoid over-speculations.

4) Related to the above comments #3, analyses of the conserved motifs in TAAR enhancers are mandatory. Are there any conserved sequence motifs in the TAAR enhancer sequences (e.g., Lhx2-binding motif, Ebf-binding(O/E-like) motif, and the Greek island composite motif)?

In addition, both the local cis-acting OR enhancers and the formation of interchromosomal enhancer hubs are required for the stable and high level expression of a single OR gene. Deletion of the cis-element (H, P, Lipsi, for example) resulted in abolishment of the liked OR genes expression and the formation of enhancer hub on a expressed OR gene locus, which are facilitated by the dimerization ability of Ldb1. At the same time, Ldb1 is required for the transcriptional activity of Lhx2, as Lhx2 do not function as a transcription factor without LIM-binding proteins. Thus, it is also possible that the reduced expression of TAAR genes in Ldb1 KO is caused by the loss of transcriptional activity Lhx2 function (local cis-acting activity) as well as failure of making up enhancer hubs.

We analyzed the conserved motifs including Ldb1-binding motif, Lhx2-binding motif, Ebf-binding motif, and Lhx2/Ebf composite motif in the two TAAR enhancers using the published Lhx2 and Ebf mOSN ChIP-seq data (Monahan et al., 2017, eLife). In TAAR enhancer 1, we identified two Ldb1- or Lhx2-binding motifs and one Lhx2/Ebf composite motif. In TAAR enhancer 2, we identified two Ldb1- or Lhx2-binding motifs, one Ebf-binding motif, and one Lhx2/Ebf composite motif (Supplementary Figure 3B, 3C). Therefore, it is possible that the two TAAR enhancers form enhancer hub together with OR enhancers in TAAR OSNs (see our response to major point 3).

Also *Lhx2* may be involved in transcriptional activation of *Taar* genes. As a result, the reduced expression of *Taar* genes in *Ldb1* knockout mice is possibly caused by failure of enhancer hub formation or loss of *Lhx2* transcriptional activity. We addressed this point in page 12 line 5-13 of the revised manuscript.

5) page 14 - 17, the description of the results of TAAR enhancer 1 knockout and enhancer 2 knockout is confusing.

For example, the authors say "the numbers of TAAR OSNs expressing *Taar3*, *Taar4*, *Taar5*...were dramatically decreased in TAAR enhancer 2 knockout mice" in page 16, line 17-19, but in the next page, "cells expressing *Taar3* and *Taar5* were completely abolished in TAAR enhancer 2 knockout mice" (page 17, line 1-2).

In page 14, line 18 "it tended to decrease for *Taar7f* and to increase for *Taar7e* (in TAAR enhancer 1 KO)." but in page 17, line 11-12 " (*Taar7f*) were not changed in TAAR enhancer 1 knockout mice."

Please rewrite these sections to ensure that there are no inconsistencies between the descriptions of the results. Also, please be consistent in the wording (there are so many terms, significantly, largely, dramatically, completely, tended to, not changed...).

In addition, please present a table summarizing the results of the RNA-seq and ISH analyses (expression level, fold of change of expression level, the number of cells, fold of change of the number, and their statistical data of each *Taar* gene in TAAR enhancer 1 KO, TAAR enhancer 2 KO, and double KO) to compare the effects of the deletion of enhancer(s) on *Taar* gene expression.

We rewrote the sections describing TAAR enhancer knockout experiments in page 14 line 6-20, page 17 line 6-14, page 19 line 16-22, and page 20 line 1-3 using consistent words. In addition, we provided Supplementary Table 2, 3, 4, 5, 6, 7, and 8 summarizing the results of RNA-seq and *in situ* experiments. In addition, the raw data were provided as source data file.

6) In Figure 6, the RNA-seq data and quantification data of apoptotic cells are missing. Please provide them. The RNA-seq data is necessary to demonstrate "no effects" on the expression of OR genes and other genes outside the cluster. The number of apoptotic cells should be presented to examine and discuss the arrest of neuronal differentiation of TAAR-OSNs.

We performed RNA-seq and apoptotic cell staining experiments on TAAR enhancer 1 & 2 double knockout mice. The data were included in Figure 6 and Supplementary Figure 7. We did not observe significant changes of OR genes in TAAR enhancer 1 & 2 double knockout mice. The number of apoptotic cells was similar in wild type, heterozygous, and homozygous TAAR enhancer 1 & 2 double knockout mice. The results were described in page 19 line 16-22, page 20 line 1-8, and page 20 line 16-20 of the revised manuscript.

7) page 19, ISH analyses of the coexpression of reporter genes and OR genes. The authors used OR mixed probes. Please describe the character of OR genes used in the mixed probe, e.g., class I or class II, dorsal or ventral OR. In this assay, the authors argued specific expression of reporter genes in TAAR-OSNs judged by

coexpression rate. However, the mixed probes are composed of 8 OR genes out of 1,100 OR genes (probably the authors chose dorsal OR genes, if so, out of ~250 genes). Considering the total number of OR genes, I do not think the authors can simply conclude the specific expression of reporter genes in TAAR-OSNs. RNA-seq analyses of FACS sorted OSNs provide convincing conclusions.

In addition, I do not understand the reason why coexpression rates of OR degenerate primers were much lower than those of OR mixed probe. Please explain. Does the degenerate probe detect class I OR genes? As the OR mixed probe includes some class I OR probes, is there any possibility that TAAR-enhancer reporters coexpress class I OR with high frequency?

The mixed *OR* probes were composed of probes recognizing 2 class I OR, 6 class II *OR* genes. They may also cross-hybridize with 6 other *OR* genes (Supplementary Table 9). Thus, the mixed OR probes we used could detect 14 out of 1,100 functional *OR* genes.

The degenerate primers used for making degenerate *OR* probe were designed following previous publications (Malnic et al., 1999, Cell; Johnson et al., 2012, PNAS). It is likely that each batch of degenerate *OR* probe contained different combinations of *OR* genes due to PCR variations. When we performed *in situ* analysis using our degenerate *OR* probe, we observed majority of positive cells in the ventral MOE. Thus, we think that our degenerate probe may barely detect class I *OR* genes.

To test the possibility that TAAR enhancer-driven reporters are largely co-expressed with class I *OR* genes, we generated the mixed class I *OR* probes composed of 8 class I *OR* probes (Iwata et al., 2017, Nat Commun), and analyzed the coexpression rates of reporters and mixed class I *OR* probes. We found that the coexpression rate was 0.4% and 1.5% in reporter-positive cells for TAAR enhancer 1-GFP and TAAR enhancer 2-tdTomato mice, respectively.

We agree with the reviewer that it is impossible to completely exclude coexpression of *OR* genes with reporter genes. Therefore, we performed RNA-seq analyses on FACS-sorted reporter-positive cells from transgenic mice, and used reporter-negative cells as controls (Supplementary Figure 9A, 9B). We found that the reporter-positive cells were indeed enriched in *Taar* genes compared to reporter-negative cells. We also investigated the coexpression rates of reporters with individual *Taar* genes (Supplementary Figure 9C, also see our response to Minor Point 7). We observed high coexpression rates for individual *Taar* genes with reporters. The sum of coexpression rates was very close to those obtained by the mixed *Taar* probes (Figure 7C, 7E, Supplementary Figure 9C).

The description of new data mentioned above was added in “**TAAR enhancers are sufficient to drive reporter expression in the TAAR OSNs**” section of the revised manuscript. All together, those data strongly suggest that the reporter-positive cells are mainly TAAR OSNs in the TAAR enhancer transgenic mice.

8) page 21, line 5-10 and other, similar expression levels (number of cells?) of Taar genes in heterozygous TAAR enhancer 1, 2 and 1&2 knockout mice compared to their wild type littermates. However, authors only provided the normalized data. One cannot evaluate the total number of cells from the ratio between knockout and control of each Taar gene. For example, in the manuscript the number of Taar3 expressing OSNs significantly decreased in TAAR enhancer 1 and enhancer 3 knockout mice. If Taar3 expressing OSNs are dominant in the TAAR-OSN population, that largely affects the total number of TAAR OSNs. Please provide all quantification data (the number of cells) as supplementary table together with statistical analyses and compare the total number of TAAR-positive cells.

We provided Supplementary Table 2, 4, and 6 summarizing the results *in situ* experiments, including the number of sections analyzed, the average number of positive cells per section, percentage of expression, and statistics. In addition, the raw data were provided as source data file.

Minor points:

1) Regarding to figure representations, please take into account the color barrier. Please replace red color with magenta or color blind friendly one.

We replaced the images in red color with those in magenta. Thanks for the kind thoughts.

2) page 7, line 19-20, "approximately 70-80% of which are composed of OR-expressing mature OSNs". How did the authors estimate the population? Please display the data, or put appropriate references.

We rephrased the sentence to "The GFP- or ZsGreen-negative cells were also sorted to serve as control cells, approximately 70-80% of which are composed of OR-expressing OSNs. This percentage was estimated from our previous observation (Tan et al., 2015, MSB).", and added the appropriate references.

3) page 12, line 10-11, "loss of TAAR enhancer 2 in dog, sea lion, seal,,,,". The VISTA plot data is missing. Please display all data described in the manuscript.

As suggested by the reviewer, we added the missing VISA plot data of dog, sea lion, and seal in Supplementary Figure 4B. We also changed the figure legend accordingly.

4) page 14, line 11-14, " No obvious differences in the numbers of cells expressing Taar7s were detected....". I think that ISH data using Taar7 coding probes is neither conclusive nor necessary (especially the expression level of Taar7e increased in the TAAR enhancer 1 KO). Please delete them from the manuscript and revise Figure 4E, 4F, 5E, 5F. The data acquired using specific probes are sufficient.

We deleted the *in situ* data using *Taar7* coding probe in Figure 4E, 4F, 5E, and 5F. We also revised the manuscript to delete the relevant text.

5) page 18, line 7, "distinct groups of Taar genes" would mislead the readers. Some Taar genes expression were decreased in both enhancer KO mice. the effects of TAAR enhancers overlap each other. Please reword.

We changed "distinct groups of *Taar* genes" to "different subgroups of *Taar* genes" in page 19 line 10 of the revised manuscript.

6) The section of "TAAR enhancers are sufficient to drive reporter expression in the TAAR OSNs".

In page 19, line 12, it is my personal opinion that it is not appropriate to include "data not shown". This is an interesting and experimentally important data, which in my opinion is well worth presenting. Because some Taar genes are expressed ventrally. Also please consider displaying both the MOE (lateral view) and the OB (both lateral and top) .

In addition, Figure 7G (the left panel) is difficult to see. Please present both single color and merge images.

We provided images of the MOE sections from TAAR enhancer 1-GFP or TAAR enhancer 2-tdTomato mice (Supplementary Figure 8D) to show that reporter-positive cells were located in both of the dorsal and ventral MOE. We also provided whole-mount fluorescent images of the lateral (Supplementary Figure 8D) and dorsal (Figure 7G) view of olfactory bulb. We did not show whole-mount fluorescent images of the lateral view of the MOE due to the high autofluorescence background in our experimental setup.

We presented the single color and merged images in Figure 7G according to the reviewer's suggestion.

7) page 20, line 14, "to examine the coordination between two TAAR enhancers". There is no doubt that endogenous TAAR enhancers coordinate to regulate Taar genes in cis. However, in the transgenic line, TAAR enhancers-GFP/tdTomato transgenes are integrated randomly into chromosomes (different position, different copy number). Apparently, two TAAR enhancers "cooperate" to regulate Taar7d and 7e, because the deletion of each enhancer did not change these expression but the deletion of both abolished them. How the authors examine the coordination between two differentially integrated transgenes? Do GFP single or tdTomato single positive OSNs reflect that only one of two endogenous enhancers is active in them? I do not think the conclusion can be drawn from the results of this experiment. Coexpression rate of each Taar genes with reports would be more helpful.

We agree with the reviewer that the TAAR enhancers in transgenes do not reflect the activities of endogenous TAAR enhancers. So we rewrote the paragraph in page 24 line 2-14 of the revised manuscript to be more precise. We also investigated the coexpression rates of reporters with individual *Taar* genes (Supplementary Figure 9C). We observed high coexpression rates for individual *Taar* genes with reporters. The sum of coexpression rates was very close to those obtained by the mixed *Taar*

probes (Figure 7C, 7E, Supplementary Figure 9C), suggesting that the reporter-positive cells are mainly TAAR OSNs.

8) page 22, line 22 "thus stabilize the cell fate decision of the TAAR subsystem". I do not understand this sentence. From which experiment can this conclusion be drawn? The authors mentioned that deletion of enhancers arrest the differentiation of TAAR-OSNs. In other words, the cell fate is not destabilized by the deletion of enhancers. Please reword.

We agree with the reviewer on this point. We deleted this sentence and reworded to "In the present study, we identified two enhancers that specifically regulate probability of *Taar* gene choice in the TAAR olfactory subsystem." in page 26 line 12-13 of the revised manuscript.

9) Figure 8C. Are there significant differences between genotype 1 versus 2, genotype 1 versus 3? Please also mention and discuss the residual *Taar6* expression in genotype 4. Are there any trans-interaction between TAAR enhancer 1 and *Taar6* genes? Only *Taar6* was unchanged between genotype 1 versus 2, whereas *Taar2*, *3* and *5* were tended to decrease (or were decreased).

We followed the reviewer's suggestion to carefully re-analyze the statistical results among the 4 genotypes in Figure 8C. We chose to examine expression of the *Taar* genes that were entirely eliminated (*Taar2*, *Taar3*, and *Taar5*) or significantly reduced (*Taar6*) in homozygous TAAR enhancer 1 knockout mice (Figure 4E, 4F, 4H). Here, genotype 1 is wild type, genotype 2 is same as heterozygous TAAR enhancer 1 knockout, and genotype 3 is same as heterozygous *Taar2-9* cluster knockout. We indeed observed similar changes of those *Taar* genes in genotype 1 versus 2 and genotype 1 versus 3 compared to those in wild type versus heterozygous TAAR enhancer 1 knockout and wild type versus heterozygous *Taar2-9* cluster knockout. We rewrote the paragraph in page 25 line 12-22 and page 26 line 1-8 of the revised manuscript to better explain the data.

10) Supplementary Figures 5, 6. Only provide all DEG data (TAAR enhancer 1, 2 and 1&2 KO) as Supplementary Table.

We added all of the RNA-seq analysis data of TAAR enhancer 1, 2, and 1 & 2 double knockout mice in Supplementary Table 8.

11) This is out of the objective of the present study, but please state the effect of TAAR enhancers knockout on *Taar1* expression in brain, if possible.

Previous studies have shown low expression levels of *Taar1* and *Taar5* in different brain areas, including hypothalamus, hippocampus, and amygdala (Borowsky et al., 2001, PNAS; Lindemann et al., 2008, J Pharmacol Exp Ther; Espinoza et al., 2020, Front Mol Neurosci). To answer the reviewer's question, we tried to detect *Taar1* and *Taar5* expression in the brain sections of wild type, heterozygous, and homozygous TAAR enhancer 1 and 2 knockout mice using *in situ* hybridization. Unfortunately, we failed to find any good signals of both *Taar* genes in the brain. This is not very

surprising, as people usually used LacZ staining to indicate *Taar* expression in the knockout mice where the *Taar* gene was replaced by *LacZ*. So we did not investigate this in further details.

Nevertheless, as the reviewer pointed, the effect of TAAR enhancers on *Taar1* or *Taar5* expression in the brain is beyond the scope of our study, and would not affect any of our conclusions.

12) page43, Figure 7 legend title. Because the authors demonstrated expression patterns of both TAAR enhancer 1 and 2, the title should be "TAAR enhancers" of "TAAR enhancer 1 and enhancer 2".

We changed the Fig. 7 title to "TAAR enhancer 1 and enhancer 2 possess TAAR OSN-specific enhancer activity".

13) In Figure 7 legend (page 43, line22) "The number of GFP-positive and tdTomato-positive founders among the total analyzed lines". In the Results, the authors mentioned to keep one transgenic line for each enhancer and analyze them. I think "among the total analyzed founders/lines" is correct. Please confirm.

We apologize for the confusion. We changed the description to "The numbers of founders having GFP-positive and tdTomato-positive cells in the MOE were also indicated. We kept one founder for each enhancer transgenic line, and used them for further experiments." In Figure 7 legend. We hope that it is clearer now.

14) The Method is not sufficient. Please provide the following information (I recommend provide detailed information as Supplementary Tables)
ISH probe information (name, genbank ID, nucleotide position, or references)
plasmid information (name, Addgene#, reference) for CRISPR-Cas9 gRNA & SpCas9 templates and for PiggyBac transposon plasmid
primer sequence for Enhancer#1 or #2 knockout genotyping
Deposit ID of the metadata analyzed in this study.

Following the reviewer's suggestions, we provided Supplementary Table 9, 10, and 11 containing the detailed information of genotyping primers, *in situ* probes, and plasmid information. We also deposited the metadata in GEO and provided the information in the "Accession codes" section of Methods. The accession number is GSE163778 (<https://www.ncbi.nlm.nih.gov/geo/query/acc.cgi?acc=GSE163778>). The secure token (ulkhcsugzdmrnm) was created to allow reviewers' access.

Reviewers' Comments:

Reviewer #1:

Remarks to the Author:

I am satisfied with the changes the authors made in response to the original critiques and I am in support of publication. This is an excellent paper.

Reviewer #2:

Remarks to the Author:

The authors have fully addressed the concerns raised in the previous review and further improved the manuscript from an already strong initial submission.

by Minghong Ma

Reviewer #3:

Remarks to the Author:

The authors satisfactorily addressed most of my points. The manuscript is now in principle acceptable. I would however suggest to correct the following points before publication.

1) page 12, line 6 and 7.

Because Ldb1 has no-DNA binding activity (and no transcriptional activation activity by itself), the expression of "Ldb1-binding motif (Ldb1 and Lhx2 have almost identical binding motifs)" is incorrect. The authors should delete the term and revised the main text, methods (p38, line 4-6), Supplementary Figure 3 and its legend (p52)).

2) Primer information has been summarized in Supplementary Table 10, please remove any primer information that is doubled in the text (p36, line 3-19).

REVIEWERS' COMMENTS

We would like to thank all of the reviewers again for their constructive comments on our manuscript.

Reviewer #1 (Remarks to the Author):

I am satisfied with the changes the authors made in response to the original critiques and I am in support of publication. This is an excellent paper.

Reviewer #2 (Remarks to the Author):

The authors have fully addressed the concerns raised in the previous review and further improved the manuscript from an already strong initial submission.

by Minghong Ma

Reviewer #3 (Remarks to the Author):

The authors satisfactorily addressed most of my points. The manuscript is now in principle acceptable.

I would however suggest to correct the following points before publication.

1) page 12, line 6 and 7.

Because Ldb1 has no-DNA binding activity (and no transcriptional activation activity by itself), the expression of “Ldb1-binding motif (Ldb1 and Lhx2 have almost identical binding motifs)” is incorrect. The authors should delete the term and revised the main text, methods (p38, line 4-6), Supplementary Figure 3 and its legend (p52)).

We deleted the expression of “Ldb1-binding motif (Ldb1 and Lhx2 have almost identical binding motifs)” and replaced with the sentence of “Ldb1 could be further recruited to facilitate OR enhancer hub formation in an Lhx2-dependent manner.” in page 12 line 11-13 of the revised manuscript. We also deleted the term of “Ldb1-binding motif” in methods, Supplementary Figure 3 and its legend according to reviewer’s suggestion.

2) Primer information has been summarized in Supplementary Table 10, please remove any primer information that is doubled in the text (p36, line 3-19).

We deleted the repeated primer information in the Methods section, and added “The probe information and primers used to generate probes were summarized in Supplementary Table 9 and 10, respectively.” in page 36 line 16-17 of the revised manuscript.